# Quercetin 3,5,7,3′,4′-pentamethyl ether from *Kaempferia parviflora* directly and effectively activates human SIRT1

Mimin Zhang [1], Peng Lu [1], Tohru Terada [2,3], Miaomiao Sui [1], Haruka Furuta[1,4], Kilico Iida[1,5], Yukie Katayama[1], Yi Lu [1], Ken Okamoto [1], Michio Suzuki [1], Tomiko Asakura [1], Kentaro Shimizu [2,3], Fumihiko Hakuno [1,4], Shin-Ichiro Takahashi [1,4], Norimoto Shimada [6], Jinwei Yang [6], Tsutomu Ishikawa [6], Jin Tatsuzaki [6] & Koji Nagata [1,3✉]

Sirtuin 1 (SIRT1), an NAD+-dependent deacetylase, is a crucial regulator that produces multiple physiological benefits, such as the prevention of cancer and age-related diseases. SIRT1 is activated by sirtuin-activating compounds (STACs). Here, we report that quercetin 3,5,7,3′,4′-pentamethyl ether (KPMF-8), a natural STAC from Thai black ginger *Kaempferia parviflora*, interacts with SIRT1 directly and stimulates SIRT1 activity by enhancing the binding affinity of SIRT1 with Ac-p53 peptide, a native substrate peptide without a fluorogenic moiety. The binding affinity between SIRT1 and Ac-p53 peptide was enhanced 8.2-fold by KPMF-8 but only 1.4-fold by resveratrol. The specific binding sites of KPMF-8 to SIRT1 were mainly localized to the helix2–turn–helix3 motif in the N-terminal domain of SIRT1. Intracellular deacetylase activity in MCF-7 cells was promoted 1.7-fold by KPMF-8 supplemented in the cell medium but only 1.2-fold by resveratrol. This work reveals that KPMF-8 activates SIRT1 more effectively than resveratrol does.

[1] Department of Applied Biological Chemistry, Graduate School of Agricultural and Life Sciences, The University of Tokyo, Bunkyo-ku Tokyo, Japan. [2] Department of Biotechnology, Graduate School of Agricultural and Life Sciences, The University of Tokyo, Bunkyo-ku Tokyo, Japan. [3] Agricultural Bioinformatics Research Unit, Graduate School of Agricultural and Life Sciences, The University of Tokyo, Bunkyo-ku Tokyo, Japan. [4] Department of Animal Resource Sciences, Graduate School of Agricultural and Life Sciences, The University of Tokyo, Bunkyo-ku Tokyo, Japan. [5] Division of Food and Nutrition, Graduate School of Human Sciences, Kyoritsu Women's University, Tokyo, Japan. [6] Tokiwa Phytochemical Co. Ltd., Sakura Chiba, Japan. ✉email: aknagata@mail.ecc.u-tokyo.ac.jp

Sirtuins are homologs of the yeast silencing information regulator 2 protein (yeast Sir2) and catalyze the removal of the acetyl group of acetylated lysine residues in protein substrates, converting $NAD^+$ into 2′-O-acetyl-ADP-ribose and nicotinamide (NAM)[1]. Among the identified mammalian sirtuins (SIRT1–7), SIRT1 has been shown to have critical functions in the prevention of many age-related diseases in mice models, such as type 2 diabetes, neurodegeneration, cancer, and Alzheimer′s disease[2–5]. A molecule that can increase SIRT1 deacetylation activity shows promise for use in the treatment of multiple diseases associated with aging. Therefore, SIRT1 has drawn the most attention as the target molecule for drug design among these seven sirtuins.

The search for molecules that activate SIRT1 began more than a decade ago. Howitz et al. discovered the first generation of STACs in 2003, using the Fluor de Lys (FdL) fluorogenic peptide substrate Ac-Arg-His-Lys-Lys(Ac)-AMC (aminomethylcoumarin). The most potent first-generation STAC is resveratrol (3,5,4′-trihydroxystilbene), a natural compound present in grapes and red wines[6]. Several classes of plant polyphenols, particularly with modifications at the B ring 4′ position, such as butein, isoliquiritigenin, and quercetin, were also confirmed to stimulate SIRT1 activity[6]. Recently, some of these natural compounds, such as resveratrol and piceatannol, were reported to prolong lifespan in yeast, metazoans and mice by acting on not only SIRT1 enzyme but also other target proteins like cyclooxygenases, lipooxygenases and MAPK family proteins. However, these natural compounds were reported to have limited health benefits in animal models, which led to a debate on their specificity and bioavailability[6–8]. Since 2007, several synthetic STACs, such as SRT1720 and SRT2410, which have an imidazo[1,2-b]thiazole core and compounds with oxazolo[4,5-b]pyridine, thiazolopyridine, benzimidazole or imidazo[4,5-c]pyridine, were reported to more potently activate SIRT1 than resveratrol and other natural compounds[2]. Although preclinical proof-of-concept efficacy data have been generated for these synthetic STACs, they have attracted wide controversies, since, SRT1720, SRT2183, and SRT1460 as well as resveratrol activated SIRT1 only towards a chemically modified peptide substrate with a covalently attached fluorophore and not toward native peptide substrates[9–11]. This controversy has spanned for several years, which has urged the continuous discovery of other SIRT1 activators.

Recent structural studies have shed light on the conserved catalytic domain (CD: 240–510) and the N- and C-terminal domains (NTD: 183–230 and CTD: 641–665) of SIRT1. In addition to the conserved catalytic domain, SIRT1 activity is regulated by its NTD and CTD[12,13]. The SIRT1 NTD is reported to be responsible for the STACs binding and The SIRT1 CTD is reported to stabilize the catalytic domain (CD) and helps form the $NAD^+$ binding pocket by completing the β sheet of the Rossman fold[14–17]. The crystal structures of STAC1–SIRT1 and resveratrol–SIRT1–Ac-p53-AMC complexes (PDB: 4ZZH and 5BTR, respectively) also provide unambiguous visual proof of the direct activation of SIRT1 by small molecules. However, in the resveratrol–SIRT1–Ac-p53-AMC complex, where a SIRT1 molecule binds a substrate peptide with an AMC moiety and three resveratrol molecules, the AMC moiety of the substrate is not involved in the interaction with SIRT1 enzyme but is involved in the interaction with all of the three resveratrol molecules[14,15]. Hubbard et al. (2013) revealed that a single amino acid, E230, located in the SIRT1 NTD, is critical for the activation of the protein by all the previously reported STAC scaffolds, and hence, proposed a common mechanism of SIRT1 regulation by STACs[18]. The mechanism of SIRT1 activation by STACs still

needs further elucidation: How does a STAC activate SIRT1 toward the native peptide substrate? Does a STAC cause the conformational change of SIRT1 for activation?

In 2014, Nakata et al. reported that polymethoxyflavonoids from the Thai black ginger Kaempferia parviflora, especially quercetin 3,5,7,3′,4′-pentamethyl ether (KPMF-8), show dramatic activation of SIRT1 activity[19]. However, the legitimacy of KPMF-8 as a direct SIRT1 activator is totally unknown, as the FdL peptide substrate was used in the initial activity assay[19]. In light of skepticism about other controversial STACs, further study is required to determine the followings: (1) whether KPMF-8 directly interacts with SIRT1, (2) whether KPMF-8 activates SIRT1 towards the native substrate, and (3) whether KPMF-8 can penetrate the cell membrane and accelerate intracellular deacetylase activity within the cellular environment. These basic in vitro studies on intermolecular interactions/recognitions are necessary and will yield important results before the potential pharmaceutical role of KPMF-8 can be tested in animal models and humans. To this end, we have herein investigated the possible interaction region of SIRT1 with KPMF-8. The results indicate that the SIRT1 NTD is critical for this interaction. A native SIRT1 substrate without the fluorogenic AMC moiety (Ac-p53 peptide) promote the binding affinity of KPMF-8 to SIRT1. KPMF-8 and Ac-p53 peptide are mutually dependent because not only Ac-p53 peptide can promote binding between KPMF-8 and SIRT1, but KPMF-8 can enhance the binding affinity of SIRT1 to Ac-p53 peptide. Further, KPMF-8 binding sites are mainly located in the α2-T-α3 motif of the SIRT1 NTD. In cooperating with Ac-p53 peptide instead of itself, KPMF-8 causes an apparent conformational change of SIRT1. KPMF-8 also significantly increases intracellular deacetylase activity in MCF-7 cells.

## Results

**Confirmation of the SIRT1 stimulating activity of KPMF-8.** To confirm the stimulating activity of KPMF-8 on SIRT1 activity reported by Nakata et al. (2014), we performed catalytic reactions with recombinant full-length SIRT1 using Ac-p53-AMC peptide (Fig. 1a, b) in the presence of 2–10 μM KPMF-8 in vitro. The stimulating activity of the samples on SIRT1 enzyme was calculated as the ratio of fluorescent intensity between samples and control (analysis buffer) (Supplementary Fig. 1a). Resveratrol, which has been considered the most potent natural SIRT1 activator over the past decade, was used as a positive control. Both KPMF-8 and resveratrol stimulated SIRT1 activity efficiently in a dose-dependent manner. KPMF-8 stimulated SIRT1 deacetylase activity roughly 25-fold at a concentration of 2 μM, while resveratrol showed the same efficiency at a concentration of 100 μM, indicating KPMF-8 was about fifty times more efficient than resveratrol. These results are consistent with the previous report[19].

Moreover, we examined the importance of the NTD and E230 (a residue in the NTD) in SIRT1 for activation by KPMF-8 or resveratrol, since the domain and this residue have been reported to be critical for the SIRT1 activation by STACs. For this purpose, we used a truncated SIRT1 protein (243–510, without the NTD) (Fig. 1a) and SIRT1-E230A mutant protein. SIRT1 (243–510) construct and SIRT1-E230A mutant still displayed 76 and 95%, respectively, of the activity of full length SIRT1 protein included in the kit, respectively (Supplementary Fig. 1b). This demonstrated that the truncated SIRT1 (243–510) and SIRT1-E230A are enzymatically active. The SIRT1 activation by KPMF-8 or resveratrol was totally abolished in the SIRT1 (243–510) truncated construct and was dramatically attenuated in the SIRT1 E230A mutant (Supplementary Fig. 1c, d). These results demonstrated the importance of the NTD and the E230 residue

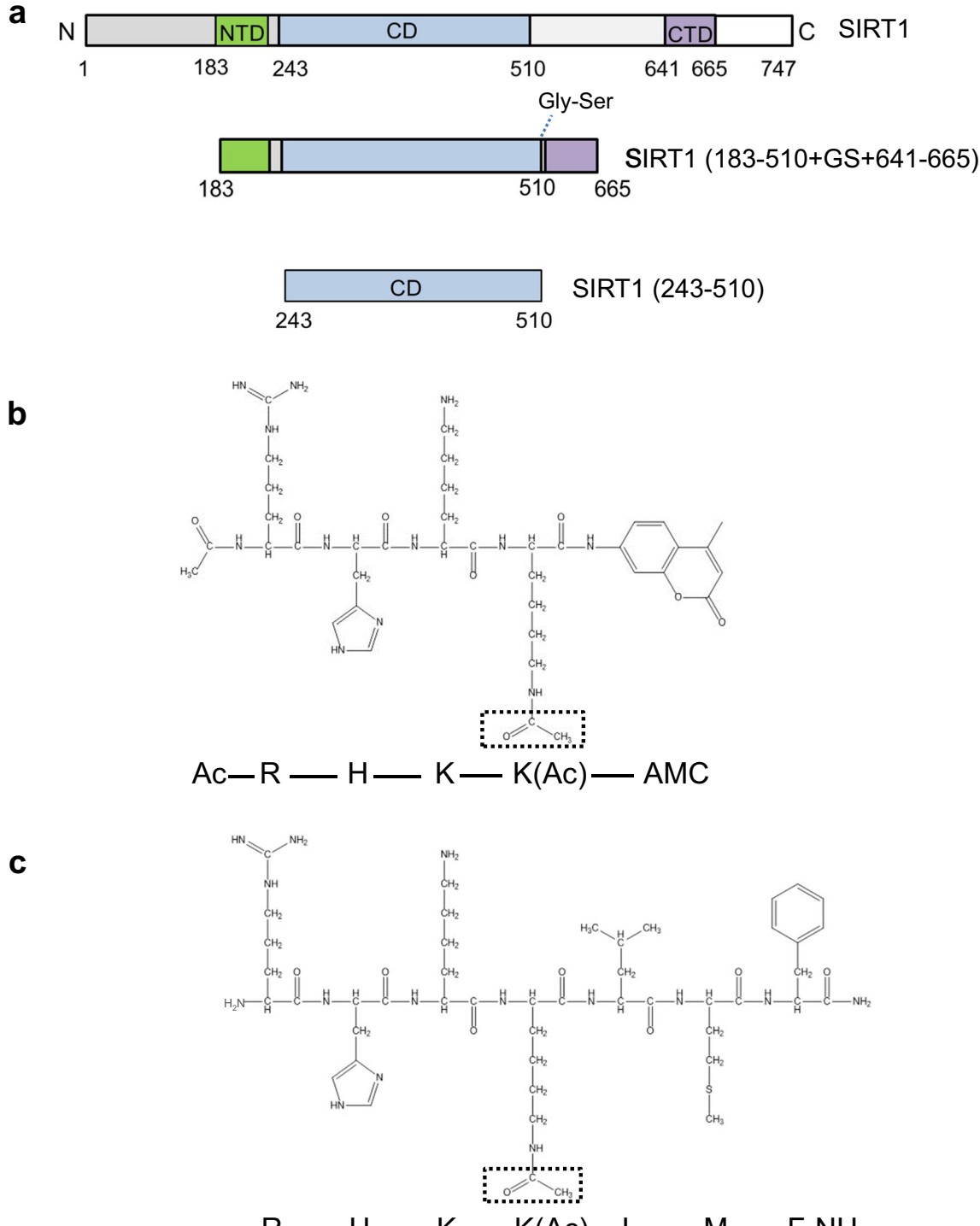

**Fig. 1 SIRT1 fragments and p53-derived peptides used in this study. a** A schematic diagram showing functional domains in full-length SIRT1 (top) and two truncation variants (SIRT1 183–510+GS+641–665 and SIRT1 243–510) used for ITC. **b** The fluorogenic peptide substrate used in the Fluor de Lys SIRT1 fluorimetric assay. **c** The native p53-derived peptide used for ITC.

for SIRT1 activation by KPMF8 as is the cases with other known STACs[14,18].

Since the substrate used in the above SIRT1 activity assay is fluorescently labeled, there is skepticism as to whether the KPMF-8 molecule is a direct SIRT1 activator and can increase SIRT1 activity towards the corresponding non-labeled peptide. To clarify this, we performed HPLC-mass spectrometry (HPLC-MS) using a non-labeled Ac-p53 peptide (Fig. 1c). This

assay uses HPLC to detect and quantify the peptide substrate and the deacetylated product peptide and subsequent ESI-MS to assign the substrate and product peaks (Supplementary Fig. 2a, b), which is based on *O*-Acetyl ADP ribose mass spectrometry (OAcADPR assay)[18]. Deacetylation was completed at 37°C in 2 h in the presence of 10 μM KPMF-8 or 10 μM resveratrol, but was incomplete in the absence of a STAC molecule (Supplementary Fig. 2c). This data showed that

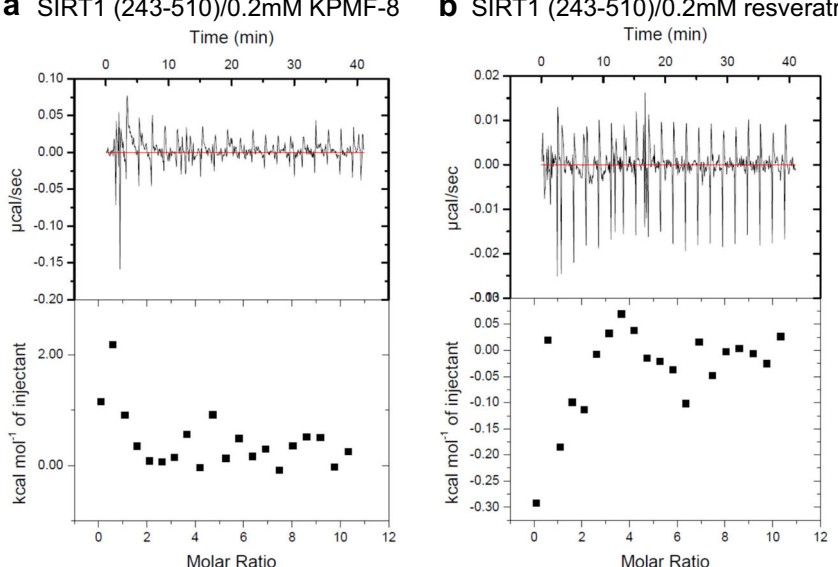

**Fig. 2 Binding of KPMF-8 or resveratrol to SIRT1 (243–510) determined using ITC. a** Binding of KPMF-8 to SIRT1 (243–510). **b** Binding of resveratrol to SIRT1 (243–510).

KPMF-8 as well as resveratrol directly activates SIRT1 towards the un-labeled substrate peptide.

**Binding analysis of KPMF-8 to SIRT1**. ITC was used to analyze the binding of KPMF-8 to different SIRT1 truncations. Resveratrol was used as a positive control in this analysis, as well. Schematic diagrams of the SIRT1 truncations are shown in Fig. 1a. The ITC data obtained by the titration of KPMF-8 or resveratrol into SIRT1 (243–510) showed little or no heat change (Fig. 2a, b), suggesting neither KPMF-8 nor resveratrol bound to SIRT1 (243–510). However, obvious heat changes were observed when titrating KPMF-8 or resveratrol into SIRT1 (183–510+GS+641–665), indicating the direct interaction of KPMF-8 or resveratrol with SIRT1 (183–510+GS+641–665) (Fig. 3a, c). The $K_D$ values, fitted using the "one set of sites" mode, suggested that KPMF-8 bound to SIRT1 more tightly than resveratrol did. A previous structural study of SIRT1 revealed that the CTD (residues 641–665) of SIRT1 appears to stabilize the CD (residues 240–510) instead of being related to STAC binding[12]. Thus, in combination with our ITC results, this allows for consideration of the possibility that SIRT1 NTD is crucial for the binding of KPMF-8 to SIRT1.

Early reports using peptide microarrays showed that the effect of activators on SIRT1 is substrate-related[20]. However, the use of fluorophore-tagged substrates in previous studies has been widely debated because STACs seem to increase SIRT1 activity toward fluorophore-tagged substrates but not toward nontagged peptides[21–23]. In 2002, Ac-p53 peptide was used as a substrate for Sir2-Af2[24]. To investigate whether the effect of KPMF-8 or resveratrol on SIRT1 activity is substrate-related, we used native Ac-p53 peptide instead of a fluorophore-tagged p53 substrate in the following experiments (amino sequence shown in Fig. 1c). In comparing the calculated $K_D$ values, we found both KPMF-8 and resveratrol bound to SIRT1 more tightly in the presence of Ac-p53 peptide (Fig. 3b, d). These data suggest that the process of SIRT1 binding to a substrate might cause a conformational change that exposes the allosteric binding site to its activator, and the mechanism of SIRT1 regulation by activators is dependent on the substrate. This finding is consistent with previous reports that SRT1460 itself did not exhibit detectable signs of SIRT1 binding but did

exhibit signs of SIRT1 binding saturation in the presence of an acetylated peptide substrate[2,18]. To rule out the possibility that KPMF-8 or resveratrol may interact with the Ac-p53 substrate, we measured the binding of KPMF-8 and resveratrol to the Ac-p53 peptide and found that neither KPMF-8 nor resveratrol showed binding (Supplementary Fig. 3a, b).

**Binding analysis of the Ac-p53 substrate to SIRT1**. Several previous studies have determined that STACs such as SRT1460 and SRT1720 activate SIRT1 by lowering its Michealis constant $(K_M)$[25,26]. However, a previous SIRT1–resveratrol–Ac-p53-AMC structure study reported no significant changes in $K_M$ values with or without resveratrol, as with a native peptide carrying no fluorophore, but the study did find that resveratrol strengthened the binding between the fluorogenic peptide and SIRT1 by serving as an adapter, which should also work in principle for native peptides[15]. To clarify the potential mechanism of SIRT1 activation by KPMF-8 or resveratrol, we analyzed the binding affinity of SIRT1 (183–510+GS+641–665) to Ac-p53 peptide with or without KPMF-8 and with or without resveratrol. For reference, no heat absorption or release was observed in the titration of Ac-p53 peptide to the buffer for SIRT1 (183–510+GS+641–665) (Fig. 4a). As shown in Fig. 4b, ITC measurement revealed that the dissociation constant $(K_D)$ between SIRT1 (183–510+GS+641–665) and Ac-p53 peptide was 55.0 μM. This binding affinity increased roughly 1.4-fold in the presence of 0.2 mM resveratrol $(K_D = 38.8$ μM; Fig. 4d). Significantly, this binding affinity increased more dramatically in the presence of 0.2 mM KPMF-8 $(K_D = 6.67$ μM; Fig. 4c), about 8-fold. These results indicate that KPMF-8 is much more efficient than resveratrol in enhancing SIRT1–Ac-p53 peptide binding, which also explains why KPMF-8 showed more activation of SIRT1 deacetylase activity (result shown in Supplementary Fig. 1a).

Taken together, these findings clearly indicate that both resveratrol and KPMF-8 stimulate SIRT1 activity by enhancing the binding affinity of SIRT1 with its Ac-p53 substrate, and KPMF-8 showed more potential in this regard.

**Binding sites of KPMF-8 or resveratrol to the SIRT1 N-terminal domain (NTD) in a solution state**. The SIRT1 NTD (residues 183–231) is necessary and sufficient for the binding of

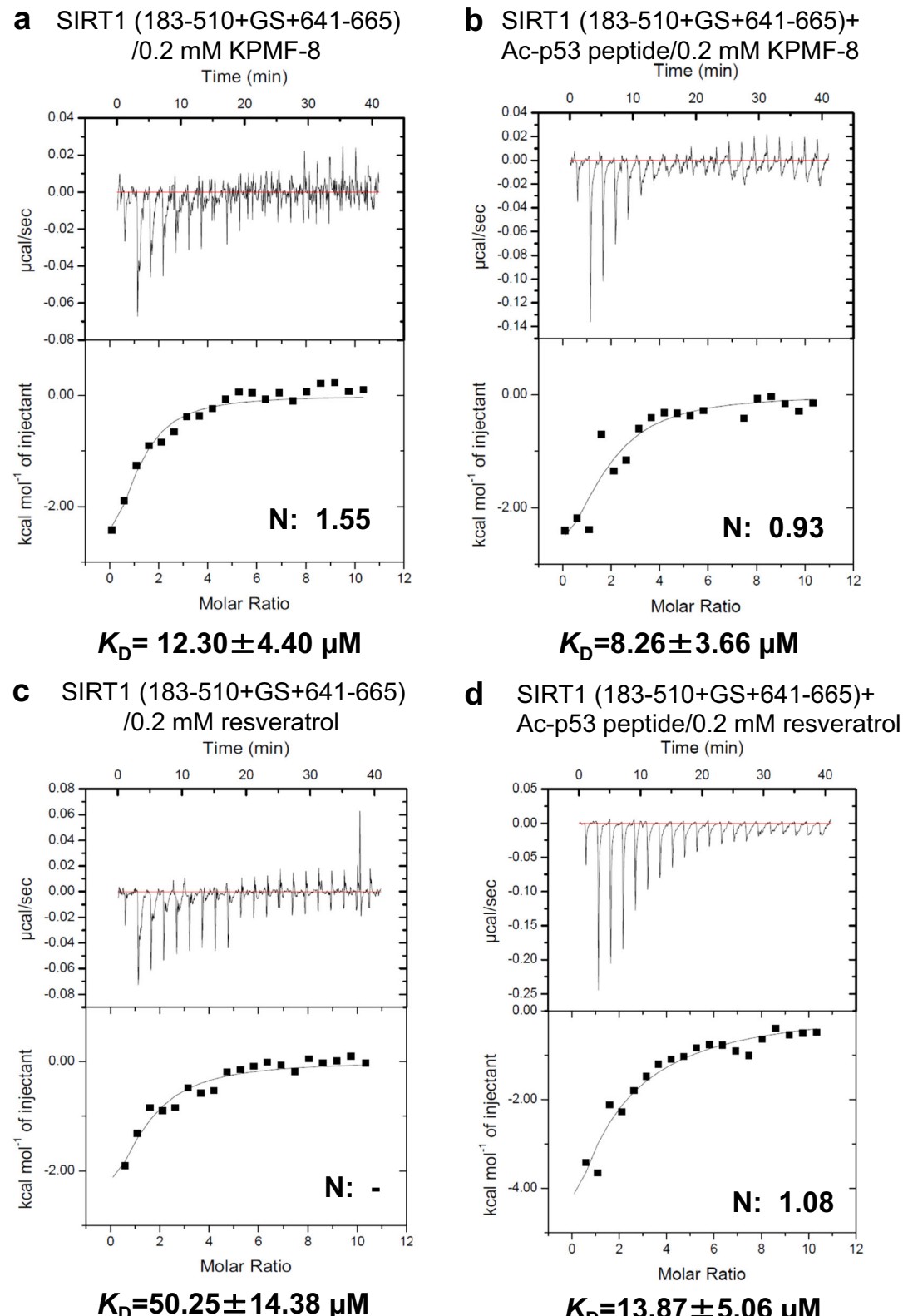

**Fig. 3 Binding of KPMF-8 or resveratrol to SIRT1 (183–510 + GS + 641–665) determined using ITC.** Binding of KPMF-8 to SIRT1 (183–510+GS+ 641–665) **a** without Ac-p53 peptide and **b** with 0.2 mM Ac-p53 peptide. Binding of resveratrol to SIRT1 (183–510+GS+641–665) **c** without Ac-p53 peptide and **d** with 0.2 mM Ac-p53 peptide. $K_D$ values ± curve-fit errors are indicated underneath each plot.

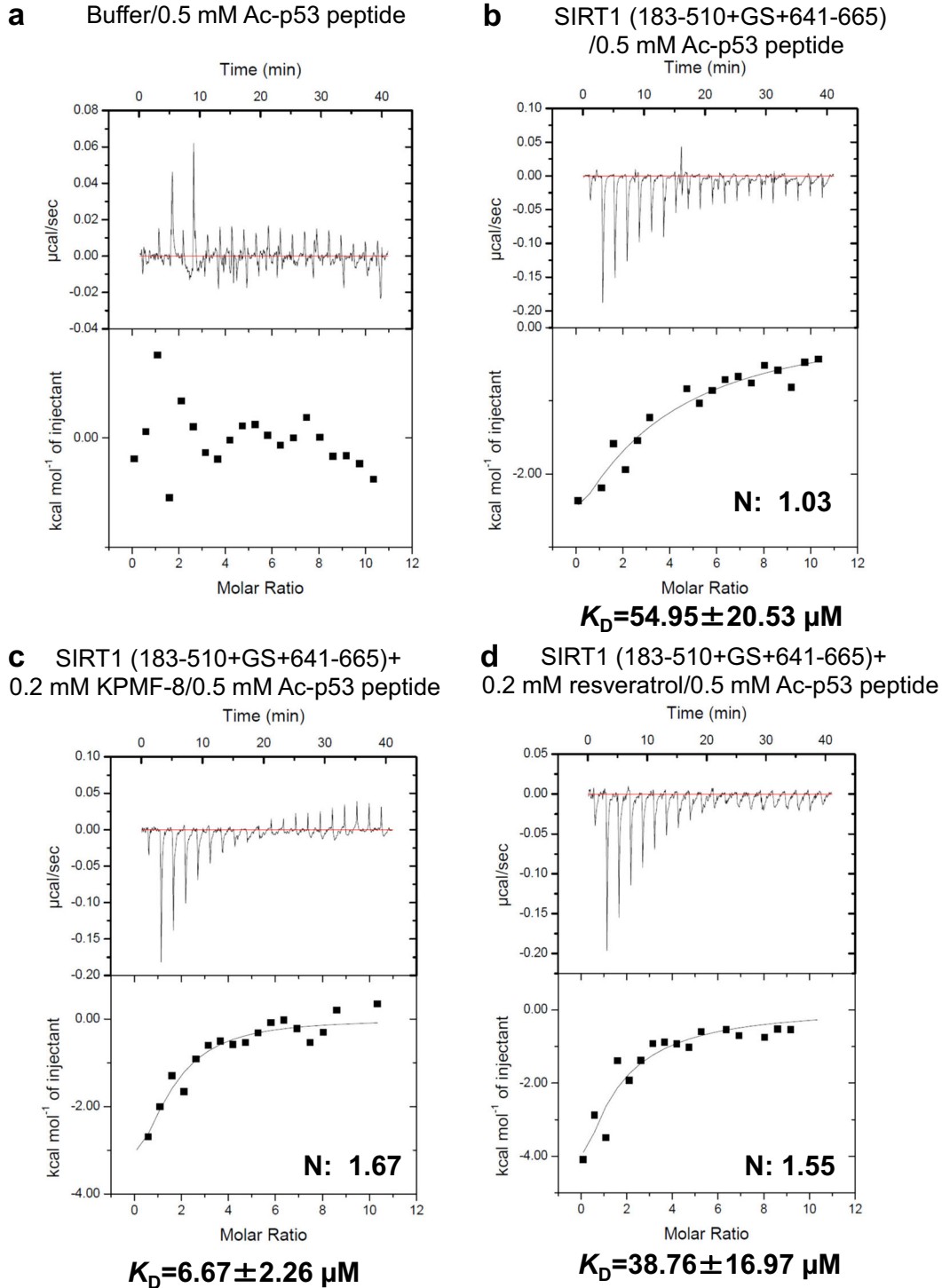

**Fig. 4 Binding of Ac-p53 peptide to SIRT1 determined using ITC. a** Titration of Ac-p53 peptide to the buffer for SIRT1 (183–510+GS+641–665). Binding of Ac-p53 peptide **b** to SIRT1 (183–510+GS+641–665), **c** to SIRT1 (183–510+GS+641–665) in the presence of 0.2 mM KPMF-8, and **d** to SIRT1 (183–510+GS+641–665) in the presence of 0.2 mM resveratrol. $K_D$ values ± curve-fit errors are indicated underneath each plot.

SIRT1 to its activator[27]. We identified the specific binding sites of KPMF-8 and resveratrol to the SIRT1 NTD using NMR spectroscopy, though the crystal structure of SIRT1 in a complex with resveratrol and Ac-p53-AMC complex (PDB: 5BTR) has already been reported[15].

The $^1$H–$^{15}$N HSQC spectrum of the SIRT1 NTD showed well-dispersed resonances, indicating a folded α-helical protein in agreement with the crystal structure of STAC1–SIRT1 complex

(PDB: 4ZZH). We assigned all the signals in the $^1$H–$^{15}$N HSQC spectra of the SIRT1 NTD (residues 183–231) alone and that supplemented with KPMF-8 or resveratrol except for three residues (Q190, L192 and I194) based on the DQF-COSY, TOCSY, and NOESY spectra as well as the chemical shift information of the SIRT1 NTD (residues 185–232) in the Bio Magnetic Resonance Bank (BMRB) database (BMRB entry: 27628)[27,28].

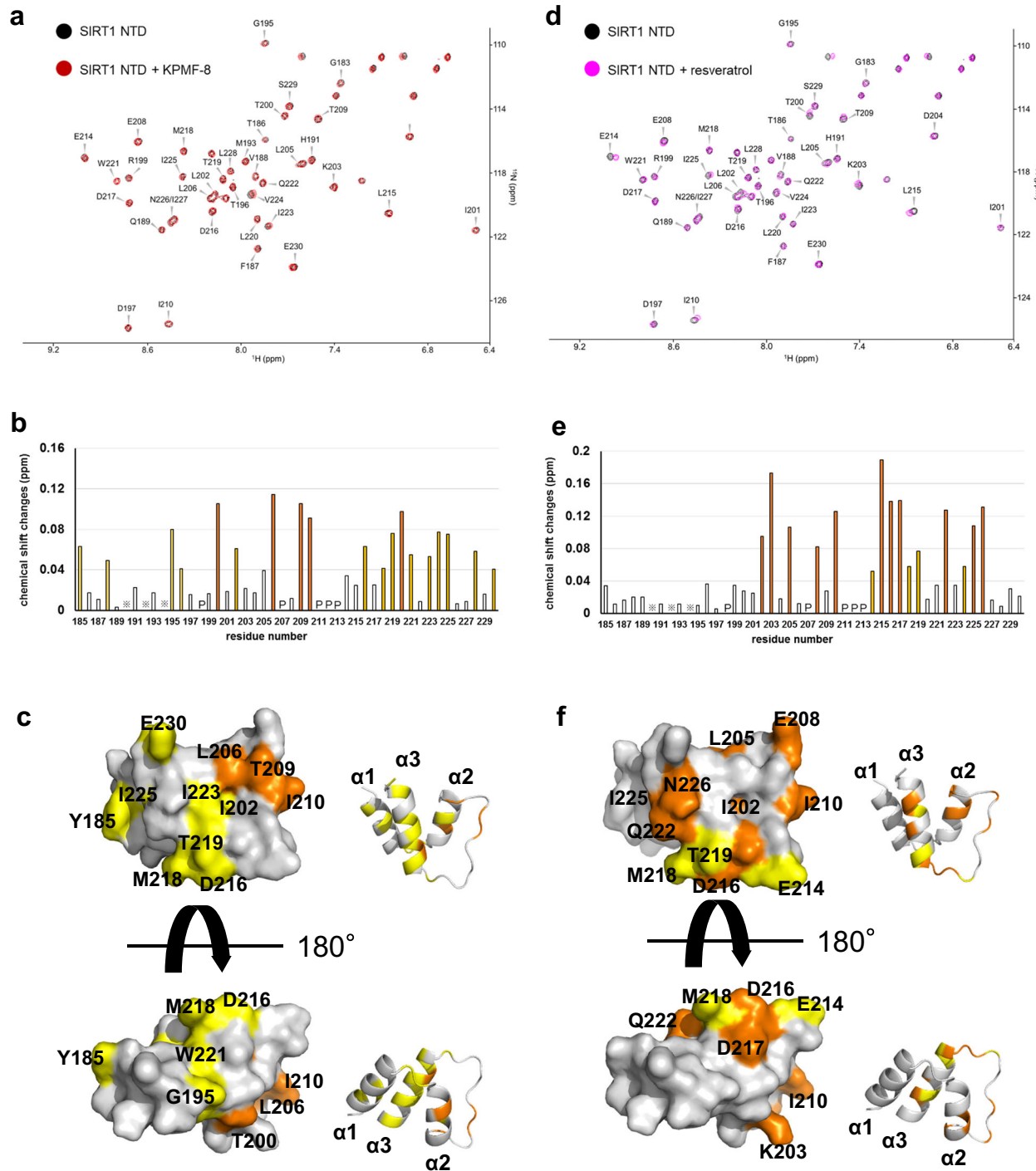

**Fig. 5 Binding of KPMF-8 or resveratrol to the 15N-labeled SIRT1 NTD analyzed by NMR.** Superposition of $^1$H-$^{15}$N HSQC spectra of 40 μM $^{15}$N-labeled SIRT1 NTD alone (black) and in the presence of **a** 400 μM KPMF-8 or **d** 400 μM resveratrol. The chemical shift changes of $^{15}$N-labeled SIRT1 NTD between the free state and the bound state to **b** KPMF-8 or **e** resveratrol. Asterisks and P's represent unassigned and unobservable proline residues, respectively. Mapping of the chemical shift-perturbed residues on the SIRT1 NTD by binding to **c** KPMF-8 or **f** resveratrol. The residues in **c** and **f** are colored according to the extent of chemical shift change defined by the formula $\delta\Delta = (\delta\Delta_{HN}^2 + 0.25 \times \delta\Delta_{15N}^2)^{1/2}$: white, 0.00–0.04 p.p.m.; yellow, 0.04–0.08 p.p.m.; orange, >0.08 p.p.m.

Comparing the $^1$H-$^{15}$N HSQC spectrum of the SIRT1 NTD with that of the SIRT1 NTD supplemented with KPMF-8 showed that a majority of signals were overlaid well, and only parts of signals had been shifted, indicating that these changes in chemical shift had been caused by specific binding of the SIRT1 NTD to KPMF-8 (Fig. 5a). Addition of resveratrol to the SIRT1 NTD solution resulted in a dramatic shift of some signals (Fig. 5d),

indicating a tight and specific interaction. To identify the specific binding sites accurately, chemical shift changes of the SIRT1 NTD caused by binding to KPMF-8 or resveratrol were evaluated quantitatively using the formula $\delta\Delta = (\delta\Delta_{HN}^2 + 0.25 \times \delta\Delta_{15N}^2)^{1/2}$ (Fig. 5b, e)[29]. In the case of KPMF-8 binding, the backbone NH resonances of T200, L206, T209, I210, and L220 exhibited large chemical shift changes (>0.08 p.p.m.), while those of Y185, V188,

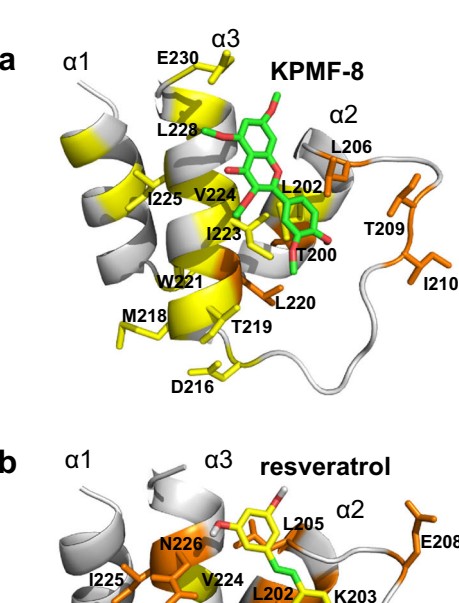

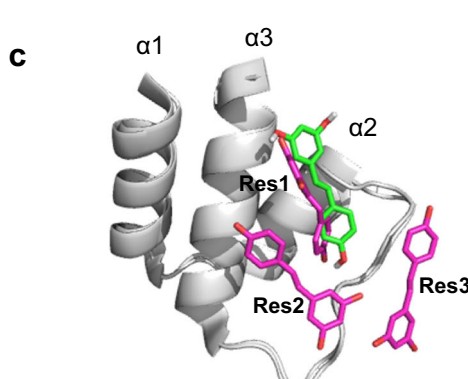

**Fig. 6 The binding model of SIRT1 NTD to KPMF-8 or resveratrol, predicted based on the structure of SIRT1 (PDB: 4ZZH) through Autodock Vina.** SIRT1 NTD residues that undergo large and moderate chemical shift changes by the addition of **a** KPMF-8 or **b** resveratrol are highlighted in orange and yellow, respectively. **c** Alignment of the resveratrol molecule docked to the SIRT1 NTD in this study (green) and the three resveratrol molecules bound to the SIRT1 NTD in the crystal structure (PDB: 5BTR) (magenta).

G195, L202, D216, M218, T219, W221, I223, I225, L228, and E230 exhibited moderate chemical shift changes (0.04–0.08 p.p.m.). In the case of resveratrol binding, the backbone NH resonances of L202, K203, L205, E208, I210, L215, D216, D217, Q222, I224, and N226 exhibited large chemical shift changes (>0.08 p.p.m.), while those of E214, M218, T219, and Val224 exhibited moderate chemical shift changes (0.04–0.08 p.p.m.).

KPMF-8 binding mainly caused chemical shift changes of the amino acid residues belonging to helix2 (α2) and helix3 (α3). The resonances of two residues belonging to helix1 (α1), Y185 and V188, as well as those of T209 and I210, which are located at the turn connecting α2 and α3, were also perturbed. Addition of KPMF-8 to the SIRT1 NTD caused pronounced chemical shift

changes in two of the four side-chain $NH_2$ resonance pairs (Fig. 5a), suggesting that KPMF-8 might also engage in asparagine and/or glutamine side-chain resonance pairs. In the case of resveratrol binding, all of the shifted residues were located in α2,α3 and the turn connecting these two α-helices.

Mapping these shifted residues on the SIRT1 NTD structure reveals that the areas with obvious chemical shift perturbations through the binding of the SIRT1 NTD to KPMF-8 and resveratrol were partially common, located mainly in the helix2–turn–helix3 (α2–T–α3) motif (Fig. 5c, f). A docking simulation was performed using Autodock Vina[30] to construct the binding models of the SIRT1 NTD with KPMF-8 and resveratrol. Based on our NMR results and the docking results, we decided to use a binding model of the SIRT1 NTD with KPMF-8 (Fig. 6a), which showed the most possible KPMF-8-SIRT1 NTD complex structure in a solution state. A docked binding model of the SIRT1 NTD and resveratrol was also constructed with the same procedure (Fig. 6b). The alignment of the docked model with the resveratrol–SIRT1–Ac-p53-AMC crystal structure showed that resveratrol was docked to almost the same position as Res1 in the crystal (Fig. 6c).

**Molecular dynamics (MD) analysis of the docked SIRT1 NTD–KPMF-8 and SIRT1 NTD–resveratrol models.** MD simulations were performed to examine the stability of the docked models. Figure 7a, d shows the root-mean-square deviation (RMSD) of the atomic coordinates from the two docked models, SIRT1 NTD–KPMF-8 and SIRT1 NTD–resveratrol, during three runs of MD simulations. In all the runs, RMSD values rapidly increased to greater than 5 Å, indicating that dissociation of the ligands from the original docked sites. However, the plots also indicated that there are several metastable states where the RMSD values fluctuated around a particular value. Cluster analysis of the MD trajectories revealed that these metastable states correspond different binding modes (Supplementary Fig. 4a, b) and exchanges between these binding modes occur. The time fractions when a ligand non-hydrogen atom was within 4 Å from a protein non-hydrogen atom were 99.96 and 99.63% for SIRT1 NTD-KPMF-8 and SIRT1 NTD-resveratrol, respectively, which indicated that both KPMF-8 and resveratrol essentially kept bound to the SIRT1 NTD.

To further investigate the interaction surface between the SIRT1 NTD and the ligands, contact probabilities of each amino acid, the percentage of time when a non-hydrogen atom of each residue was within 4 Å from a non-hydrogen atom of the ligand, was calculated based on the MD trajectories (Fig. 7b, e). Residues with high contact probabilities (>50%; orange, 50–20%; yellow) were located at the α2–T–α3 motif (G195 to P231) in both the complexes (Fig. 7c, f). Consistent with our NMR data, residues located in the α1 motif of the SIRT1 NTD–KPMF-8 complex showed relatively higher contact probabilities than those in the SIRT1 NTD–resveratrol complex, indicating that the interaction surface between KPMF-8 and SIRT1 NTD is broader than the surface between resveratrol and SIRT1 NTD.

**SIRT1 conformational change caused by KPMF-8 or resveratrol.** Previous structural studies have strengthened the hypothesis that the relationship of STACs with SIRT1 is not only complex but sometimes also ambiguous. In the SIRT1–resveratrol–Ac-p53-AMC crystal structure (PDB: 5BTR), resveratrol bridges the SIRT1 NTD and Ac-p53-AMC, making the SIRT1 conformation more compact, whereas in the SIRT1–STAC1 complex structure (PDB: 4ZZH), the SIRT1 conformation seems much less compact. Moreover, it remains unclear whether STACs themselves induce a conformational change of SIRT1, as information about the apo

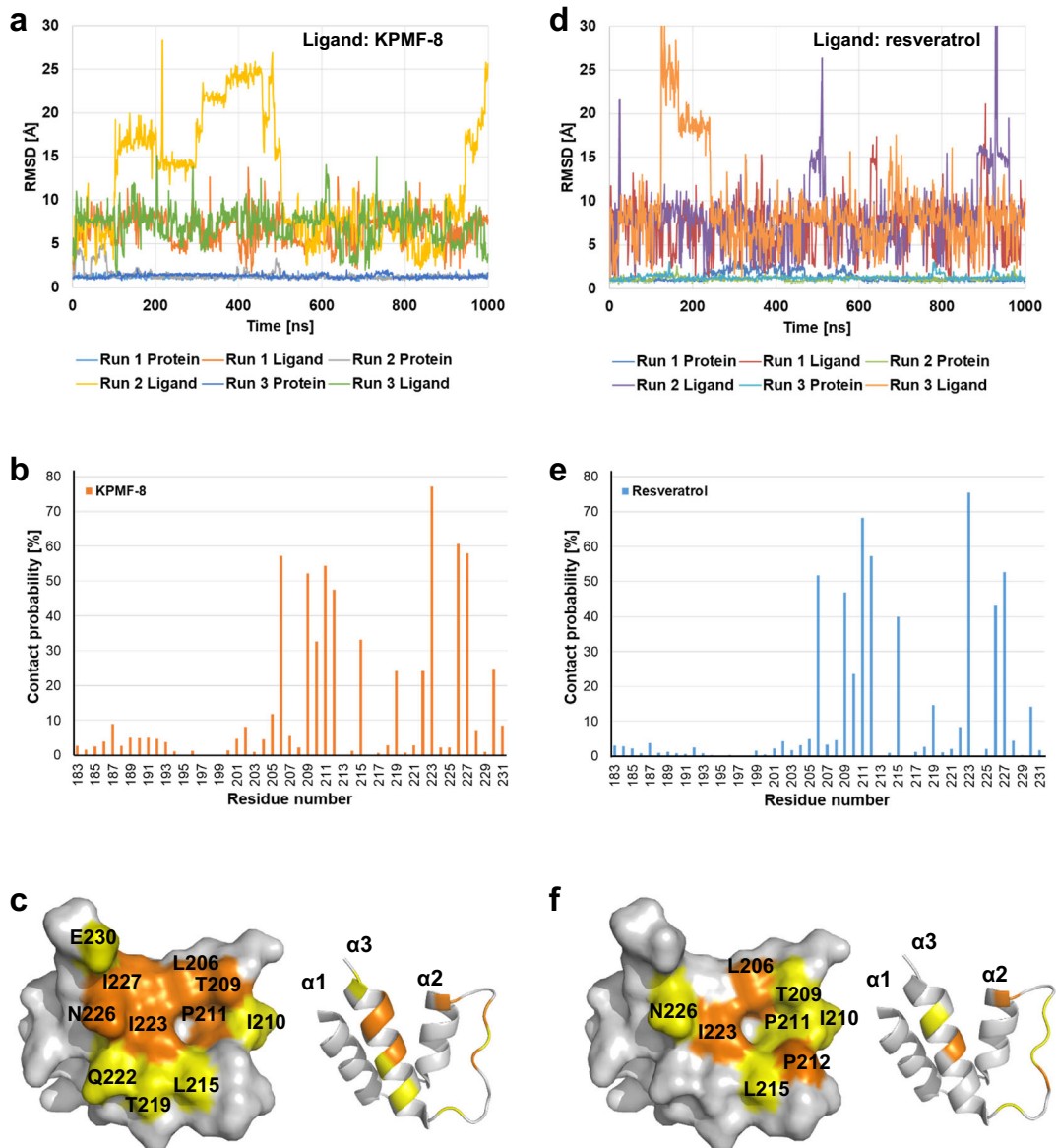

**Fig. 7 RMSD values of SIRT1 NTD–KPMF-8 and SIRT1 NTD-resveratrol models as determined by MD simulations. a** SIRT1 NTD–KPMF-8 model. **d** SIRT1 NTD-resveratrol model. RMSD values were calculated for Cα atoms of the protein and non-hydrogen atoms of the ligands using the docking models as the reference structures after aligning the Cα atoms of the protein of the MD structures to those of the docking models. Contact probabilities of each residue in **b** SIRT1 NTD–KPMF-8 model and **e** SIRT1 NTD-resveratrol model as calculated by MD simulation. Mapping of the residues with high contact probabilities on the SIRT1 NTD complexed with **c** KPMF-8 and **f** resveratrol (20–50% yellow; orange, > 50%).

SIRT1 structure is lacking. We aimed to determine the SIRT1 conformational change induced by KPMF-8 using a fluorescent resonance energy transfer (FRET) assay.

To measure the SIRT1 conformational change, we tagged Clover (a green fluorescent protein) and mRuby2 (a red fluorescent protein) to the SIRT1 N- and C-termini, respectively (schematic diagram shown in Supplementary Fig. 5)[31]. As shown in Fig. 8, the fluorescent intensity ratio of mRuby2 to Clover, Int (mRuby2)/Int (Clover), did not change among Clover–SIRT1–mRuby2, Clover–SIRT1–mRuby2 with KPMF-8, Clover–SIRT1–mRuby2 with resveratrol, and Clover–SIRT1–mRuby2 with Ac-p53 peptide. This indicated that KPMF-8, resveratrol, and Ac-p53 peptide could interact with SIRT1 without causing a detectable conformational change. We also measured the Int (mRuby2)/Int (Clover) of Clover–SIRT1–mRuby2 with both KPMF-8 and Ac-p53 peptide as well as Clover–SIRT1–mRuby2 with both resveratrol and Ac-p53 peptide. The Int (mRuby2)/Int (Clover) decreased significantly in the presence

of KPMF-8 and Ac-p53 peptide, which indicated the distance and/or arrangement of these two fluorescent proteins were/was changed. This finding suggested that the presence of both KPMF-8 and Ac-p53 peptide caused a conformational change of SIRT1. In contrast, the Int (mRuby2)/Int (Clover) did not show any significant change in the presence of resveratrol and Ac-p53 peptide, suggesting no detectable conformational change was caused in SIRT1 (Fig. 8).

**Influence of KPMF-8 and resveratrol on HDAC activity in cells**. KPMF-8 has been confirmed to have a higher potential for SIRT1 activation than resveratrol in the forepart (Supplementary Fig. 1a). However, no studies have examined the effect of KPMF-8 on deacetylase activity within a cellular environment. To investigate it, we assessed the effect of KPMF-8 on intracellular deacetylase activity in MCF-7 cells and compared its efficacy with that of resveratrol. Compared with the control, both KPMF-8 and resveratrol promoted deacetylase activity with significance in

MCF-7 cells (Fig. 9a). KPMF-8 had a significant stimulating effect on HDAC (histone deacetylase) activity, with a 1.7-fold promotion of intracellular deacetylase activity, while resveratrol showed only a minor effect, with a 1.2-fold promotion. Furthermore, we compared the expression level of SIRT1 in normal MCF-7 cells

treated with/without resveratrol or KPMF-8 and found neither of these two compounds affected the SIRT1 expression level (Fig. 9b, Supplementary Fig. 6).

To distinguish SIRT1 from other sirtuins and deacetylases in MCF-7 cells, we performed SIRT1-knockdown experiments (Fig. 9c, d). MCF-7 cells were transiently transfected with one of the three independent siRNAs against SIRT1 (1, 2 and 3) or control siRNA (NC). We confirmed that SIRT1 protein in MCF-7 cells was knocked down by all the siRNAs used in the experiment (Fig. 9d, Supplementary Fig. 6). The stimulation of deacetylase activity observed in normal MCF-7 cells was completely abolished in the SIRT1-knockdown MCF-7 cells (Fig. 9c), which showed that SIRT1 is the target deacetylase activated by KPMF-8 or resveratrol in normal MCF-7 cells. These results demonstrated that KPMF-8 and resveratrol supplemented in the medium can stimulate the deacetylase activity of SIRT1 within the cells without affecting its expression level.

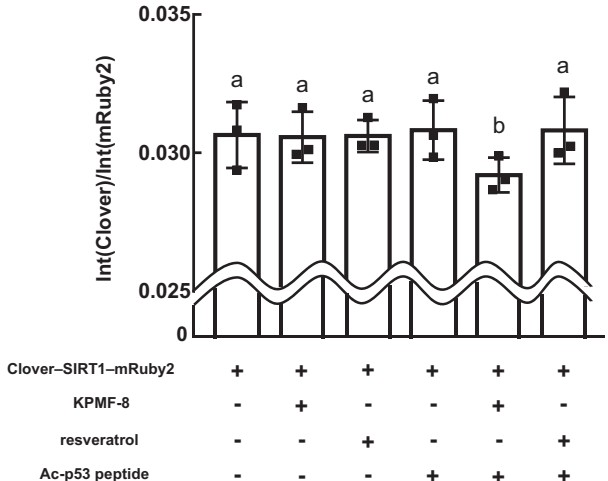

**Fig. 8 FRET measurements of Clover–SIRT1–mRuby2 in the absence or presence of KPMF-8, resveratrol and Ac-p53 peptide.** Results are presented as the fluorescent intensity ratio of mRuby2 to Clover (Mean ± SD, $n = 3$ independent experiments). Means with the same letter are not significantly different ($p < 0.05$).

## Discussion
This study identifies KPMF-8 as a promising SIRT1 activator: (1) it bound to SIRT1 directly and upregulated SIRT1 activity through enhancing the binding affinity of the enzyme with its substrate (Ac-p53 peptide); (2) binding sites between KPMF-8 and SIRT1 NTD were specified; (3) KPMF-8 cooperates with Ac-p53 peptide to cause the conformational change of SIRT1; and (4) KPMF-8, when added in the culture medium, could promote intracellular deacetylase activity.

Regarding the activation mechanism of SIRT1, researchers have adopted two views: one is that the activator binds to the substrate, and the other is that the activator binds to the enzyme.

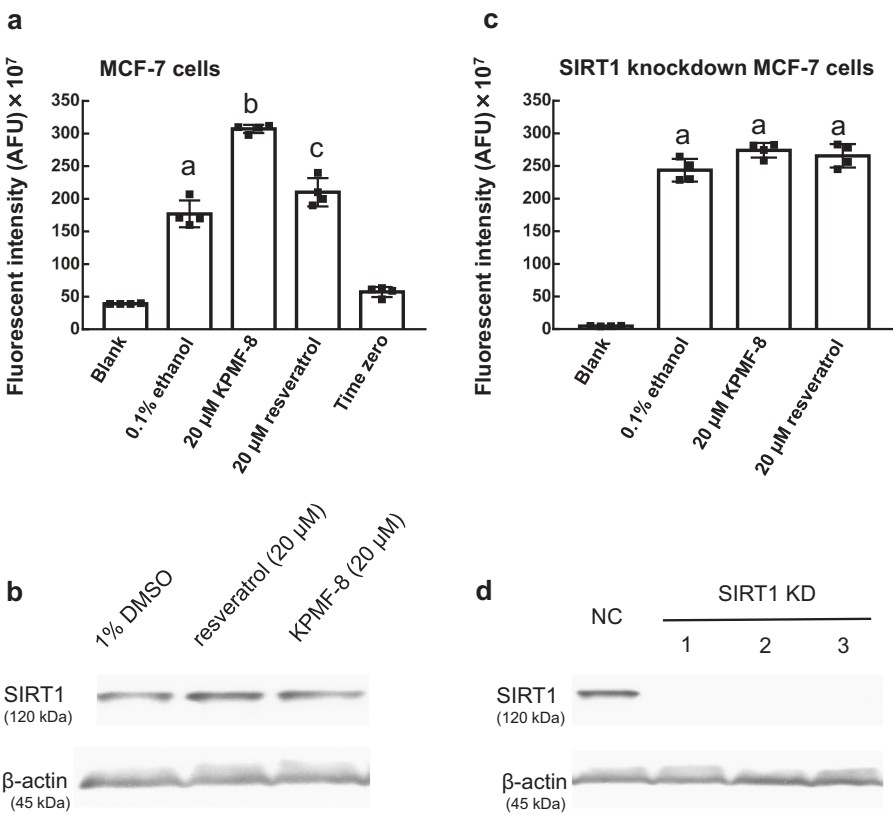

**Fig. 9 Stimulation of intracellular deacetylase activity within normal or SIRT1 knockdown MCF-7 cells by externally added KPMF-8 or resveratrol. a** normal MCF-7 cells. **c** SIRT1 knockdown MCF-7 cells. The SIRT1 knockdown MCF-7 cells in **c** were obtained after siRNA2 treatment. Results are presented as fluorescent intensities (Mean ± SD, $n = 4$ biologically independent samples). Means with the same letter are not significantly different ($p < 0.05$). **b** Western blot analysis of SIRT1 expression in normal MCF-7 cells treated by resveratrol or KPMF-8. **d** Western blot analysis of SIRT1 in control siRNA (NC) transfected MCF-7 cells and SIRT1 knockdown MCF-7 cells obtained using siRNA1, 2, and 3, which are labeled as 1, 2, and 3, respectively.

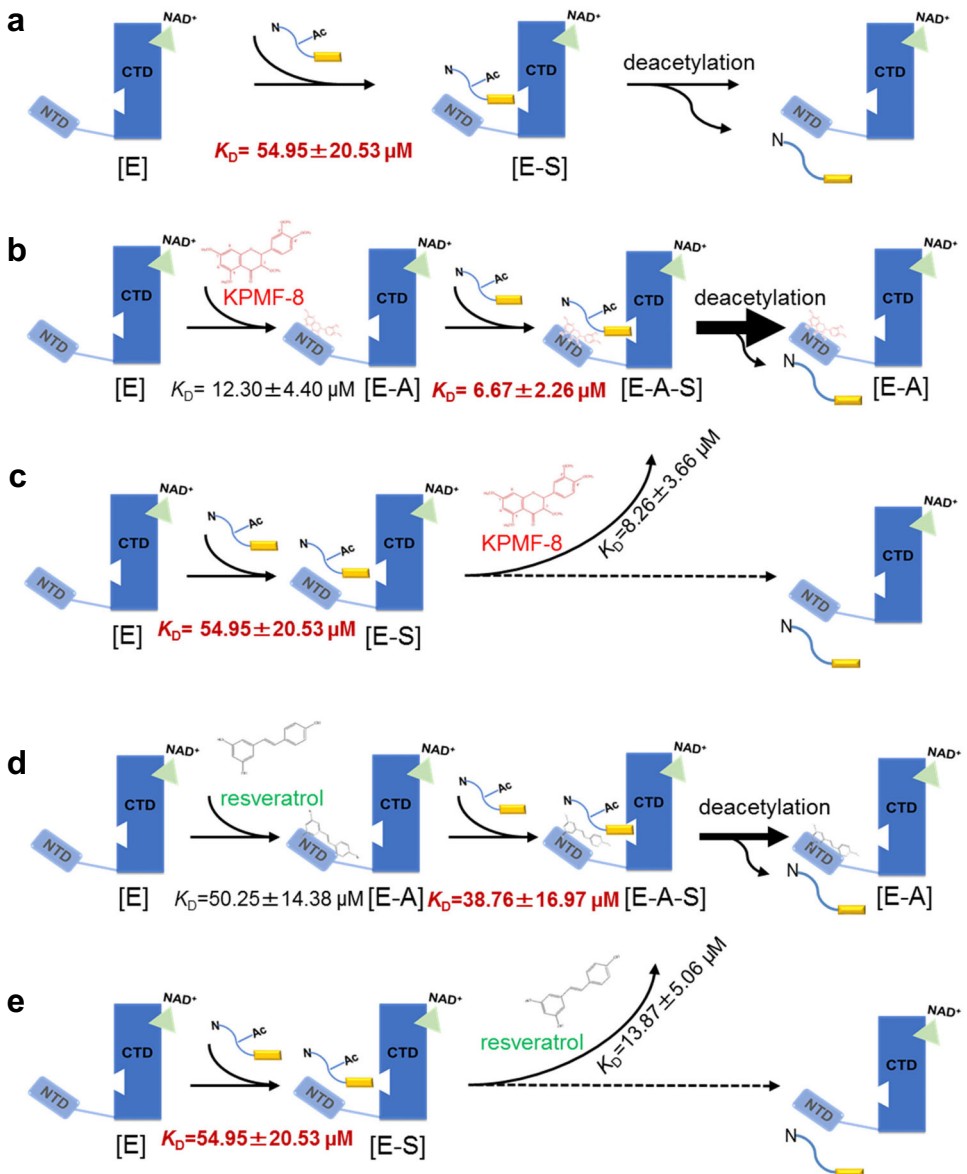

**Fig. 10 Proposed mechanism for SIRT1 activation by KPMF-8 and resveratrol. a**, **b**, **d** The binding of SIRT1 [E] and Ac-p53 peptide [S] in the absence of an activator, in the presence of KPMF-8 and in the presence of resveratrol. **b**, **c** The binding of KPMF-8 to SIRT1 [E] and SIRT1–Ac-p53 peptide [E–S] complex. **d**, **e** The binding of resveratrol to SIRT1 [E] and SIRT1–Ac-p53 peptide [E–S] complex.

Our ITC results revealed that both KPMF-8 and resveratrol bound to SIRT1 instead of to Ac-p53 peptide directly, but the presence of Ac-p53 peptide increased the binding affinity between SIRT1 and these two activators (Fig. 3).

Proposed activation mechanism in our study is the binding of KPMF-8 or resveratrol to SIRT1 assists substrate binding by lowering the $K_D$ value of SIRT1 [E] and Ac-p53 peptide [S] (Fig. 4 and Fig. 10a, b, d), which leads to an apparently increased activity of SIRT1. Moreover, both KPMF-8 and resveratrol bind more efficiently to SIRT1–Ac-p53 peptide [E–S] complex than to SIRT1 [E], as shown by the lower $K_D$ values of these activators to the [E–S] complex than to [E] only (Fig. 10b–e). These observations highlight the complex tripartite relationship between SIRT1, the activator and the substrate and are in accordance with a proposed mechanism of "assisted allosteric activation" in previous studies, which means the binding of STAC to SIRT1 NTD results in a preferred conformation for substrate binding and then increases

enzyme activity by lowering the $K_M$ for the acetylated substrate[32,33].

In addition, we specified the binding sites of KPMF-8 and resveratrol to the SIRT1 NTD in a solution state, though a resveratrol–SIRT1–Ac-p53-AMC crystal structure (PDB: 5BTR) has been solved. KPMF-8 binding sites were mainly located in the α2–T–α3 motif. Amino acid residues with large or medium chemical shift perturbations, including G195, L202, L206, I210, M218, L220, I223, I225, and L228 form a shallow hydrophobic surface suitable for interacting with KPMF-8 that has a hydrophobic nature. Residue E230, which is reported to be crucial for the activation of SIRT1 by STACs[18], showed a moderate chemical shift perturbation in our NMR data of the complex between SIRT1 NTD and KPMF-8. With regard to the complex formed with resveratrol, the ligand formed hydrophobic interactions with several residues, including I202, L205, I210, L215, M218, V224, and I225, which are also located in the α2–T–α3 motif of the

SIRT1 NTD. This finding is consistent with that of a previous study showing the binding of one of the three resveratrol molecules, Res1, in the resveratrol–SIRT1–Ac-p53-AMC crystal structure[15].

The study of the crystal structure of SIRT1 with its activator has laid the foundation for an examination of the mechanism of activation of SIRT1 by STACs. However, we still do not know how the activator binding changes SIRT1 conformation, as we lack information on the structure of apo SIRT1. Our FRET assay indicated a larger distance between the SIRT1 NTD and CTD in the KPMF–8–SIRT1–Ac-p53 peptide mixture than in apo SIRT1. This result makes it difficult to elucidate how KPMF-8 could enhance SIRT1–substrate binding affinity and upregulate SIRT1 activity. To answer this question, further structural study is required, such as an examination of apo SIRT1 and a KPMF-8–SIRT1–Ac-p53 peptide complex.

In addition to the remarkable activation of SIRT1 by KPMF-8 in non-cellular systems, we observed a promotion of intracellular deacetylase activity in MCF-7 cells, suggesting KPMF-8 is cell-permeable and can activate the intracellular SIRT1 molecules. It should be noted that there are three classes of HDACs, class I and class II enzymes, which remove acetyl groups by hydrolysis, and SIRT1, known as a class III enzyme, which employs $NAD^+$ and proceeds as a sequential biochemical reaction[7]. As such, further research is needed to determine whether this observed deacetylase activation was due to direct stimulation of SIRT1.

In summary, the findings presented in our study provide unambiguous proof of a direct interaction between KPMF-8 and SIRT1. The mechanism for SIRT1 activation by KPMF-8 and resveratrol has been proposed based on ITC, NMR, and a FRET assay. KPMF-8 showed potential as an adapter to strengthen the binding of SIRT1 to its native substrate and also worked efficiently in a cellular environment. Therefore, KPMF-8 holds promise as a new target for application in pharmaceutical trials. Further studies on the SIRT1–KPMF–8–substrate crystal structure and the confirmation of SIRT1 stimulation activity by KPMF-8 in vivo are worthy of being carried out.

## Methods

**Materials**. KPMF-8 was synthesized from quercetin[34]. Resveratrol was purchased from Fujifilm Wako Pure Chemical (Osaka, Japan). $^{15}N$-labeled $NH_4Cl$ was purchased from Cambridge Isotope Laboratories (Tewksbury, MA, USA). Native Ac-p53 peptide was purchased from DG peptide Co., Ltd (Hangzhou, China).

**Assay of SIRT1 enzyme-stimulating activity**. The FdL SIRT1 fluorimetric drug discovery assay kit (BML-AK555-0001, Enzo Life Sciences, Farmingdale, NY, USA) was used to measure SIRT1 activity according to the manufacturer's instructions. The stimulating activity of the samples on the SIRT1 enzyme was calculated as the ratio of fluorescent intensity between samples and control, with fluorescence excitation at 360 nm and emission at 460 nm.

**Protein cloning, expression and purification**. SIRT1 (243–510) and SIRT1 (183–510+GS+641–665) constructs were cloned into a pET47b vector (Novagen) at the restriction enzyme site of BamHI. The proteins were expressed in Escherichia coli BL21 (DE3) cells (Novagen) as an N-terminal fusion to a hexa-histidine ($His_6$) affinity tag with a human rhinovirus 3 C (HRV3C) protease cleavage site. A single colony was inoculated in 10 ml LB media containing 20 ug/ml kanamycin at 37 °C, 200 r.p.m., for 5 h. The pre-culture was then transferred to 1 L LB media, at 105 r.p.m., until the $A_{600}$ reached 0.6–0.8. Isopropyl 1-thio-β-D-galactopyranoside (IPTG) was added to a final concentration of 0.5 mM, and expression was continued at 16 °C, 100 r.p.m., overnight.

Cells were collected by centrifugation, and the pellet was resuspended in a lysis buffer of 20 mM Tris-HCl, pH 8.0, 200 mM NaCl, 20 mM imidazole, 1% glycerol, and 1 mM tris(2-carboxyethyl)phosphine (TCEP) and sonicated to break the cells. The supernatant was separated from cell debris by centrifugation at 40,000 × g for 30 min at 4 °C and loaded onto a Ni-NTA column (Qiagen, Germantown, MD, USA) that equilibrated with the lysis buffer. The column was washed with ten column volumes of a buffer containing 20 mM Tris-HCl, pH 8.0, 200 mM NaCl, 40 mM imidazole, 1% glycerol, and 1 mM TCEP and eluted with a buffer containing 20 mM Tris-HCl, pH 8.0, 200 mM NaCl, 250 mM imidazole, 1% glycerol, and 1 mM TCEP. The eluted protein was dialyzed in a lysis buffer and

digested with HRV3C protease to remove the N-terminal $His_6$ tag at 4 °C overnight.

The protein was diluted with a buffer containing 20 mM Tris-HCl, pH 8.0, and 1 mM TCEP to lower the concentration of NaCl and was further purified with a Mono Q column (GE Healthcare, Chicago, IL, USA). Elution was performed with a linear gradient of 0–1.0 M NaCl in a buffer of 20 mM Tris-HCl, pH 8.0, and 1 mM TCEP. The target protein was further applied to a HiLoad 10/300 Superdex 200 column (GE Healthcare) to remove protein impurities. The running buffer contained 20 mM HEPES, pH 7.5, 200 mM NaCl, and 1 mM TCEP. The purity of the target protein was confirmed by SDS-polyacrylamide gel electrophoresis (SDS-PAGE) analysis.

A SIRT1 NTD (183–231) construct was inserted into pET32a (Novagen) with an N-terminal thioredoxin tag (Trx-tag) and hexa-histidine tag ($His_6$ tag) and a Tobacco Etch Virus (TEV) protease cleavage site. The verified recombinant plasmid was used to transform to E. coli BL21(DE3) (Novagen). A freshly transformed colony was transferred to 10 mL LB medium containing 20 μg mL$^{-1}$ ampicillin antibiotic and cultured at 37 °C for 5 h. The 10 mL culture was transferred into 1 L LB medium and was grown at 37 °C until $OD_{600}$ reached 0.8. To prepare the $^{15}N$-labeled SIRT1 NTD, the cells were collected by centrifugation at 1000 × g for 15 min, resuspended with 100 mL sterilized M9 medium containing $^{15}NH_4Cl$, transferred to a large-scale sterilized M9 medium containing $^{15}NH_4Cl$ (1 L), and grown at 18 °C until $OD_{600}$ reached 0.8 again. Overexpression of the SIRT1 NTD was induced by adding 0.5 mM IPTG (final concentration) at 18 °C overnight.

The cells were harvested by centrifugation, and the pellet was resuspended in a lysis buffer (20 mM Tris-HCl, pH 8.0, 200 mM NaCl, 20 mM imidazole). The resuspended cells were lysed by sonication, and cell debris was then removed by centrifugation at 40,000 × g for 30 min at 4 °C. The supernatant fraction of the SIRT1 NTD was then applied to a Ni-NTA agarose resin (Qiagen) pre-equilibrated with a lysis buffer. The column was washed with ten column volumes of a buffer containing 20 mM Tris-HCl, pH 8.0, 200 mM NaCl, and 40 mM imidazole and eluted with a buffer containing 20 mM Tris-HCl, pH 8.0, 200 mM NaCl, and 300 mM imidazole. The elution was diluted with a buffer containing 20 mM Tris-HCl, pH 8.0, to lower the concentration of NaCl and was further purified with a Mono Q column (GE Healthcare). Elution was performed with a linear gradient of 0–1.0 M NaCl in a buffer of 20 mM Tris-HCl, pH 8.0. TEV protease was added to the elution containing the target protein at 4 °C overnight to remove the Trx-His6-tag, and the untagged SIRT1 NTD was purified by gel filtration with a HiLoad 10/30 Superdex 75 column (GE Healthcare) to remove protein impurities. A buffer containing 20 mM HEPES, pH 7.5, 100 mM NaCl was used as a running buffer. The purity of the SIRT1 NTD was confirmed by Tricine-SDS-PAGE.

**HPLC-mass spectrometry assay**. To generate p53 peptide, 0.1 mM Ac-53 peptide and 0.2 mM $NAD^+$ was incubated with 10 μM SIRT1 in 200 mL of buffer containing 50 mM Tris-HCl (pH 7.4) and 150 mM NaCl for 2 h at 37 °C, before analysis of products by RP-HPLC on a PEGASIL ODS SP100 (4.6 ×250 mm) at room temperature. An isocratic elution (1.0 ml/min) with 0.05% trifluoroacetic acid in water for 5 min was followed by a linear gradient of 0.05% trifluoroacetic acid in water to 0.05% trifluoroacetic acid in 60% acetonitrile/40% water from 5 to 45 min. UV absorbance at 220 nm was detected. Ac-p53 peptide was eluted at 18 min, produced p53 peptide was eluted at 17 min. Formic acid was added to collected samples to a final concentration of 1% prior to direct infusion into the MS system at a flow rate of 0.5 ml/min. All ESI-MS spectra were obtained in positive ion mode with an electrospray capillary potential of 2.0 kV.

**ITC assay**. Binding assays were performed using a MicroCal iTC200 isothermal titration calorimeter (GE Healthcare) at 25 °C. Protein concentration was adjusted to a final figure of 10 μM, with the assay buffer consisting of 20 mM HEPES, pH 7.4, 150 mM NaCl, and 0.5% ethanol. KMPF-8 (200 μM), resveratrol (200 μM), and native Ac-p53 peptide (500 μM) were prepared using the assay buffer. The reference power was set at 10 μCal/sec, the initial delay was set at 120 s, and the sample cell was stirred at 1000 r.p.m. A total of 20 injections, with a 0.4-μL first drop and 2.0-μL subsequent drops, was titrated into a 200-μL well. Buffer control for each experiment was performed under the same conditions, and background heat, measured by titrating the compound or native Ac-p53 peptide into the buffer in the same manner, was subtracted from the integrated data. All of the data were integrated and fitted with a one-site model using the Origin software provided by GE Healthcare.

**NMR measurements**. All the NMR spectra were measured at 10 °C on a 500 MHz Unity INOVA spectrometer (Agilent Technologies, Santa Clara, CA, USA). NMR samples of the $^{15}N$-labeled SIRT1 NTD (final concentration 0.04 mM), the $^{15}N$-labeled SIRT1 NTD with KPMF-8 (final concentrations 0.04 mM and 0.4 mM, respectively), and the $^{15}N$-labeled SIRT1 NTD with resveratrol (final concentrations 0.04 mM and 0.4 mM, respectively) were prepared in 20 mM HEPES, pH 7.5, 100 mM NaCl, and 10% perdeuterated ethanol.

$^1H$ − $^{15}N$ HSQC data as well as $^1H$ − $^1H$ DQF-COSY, TOCSY and NOESY data were recorded at 20 °C. All $^1H$ spectra were referenced to DSS at 0 p.p.m. Spectra were processed with NMRPipe and analyzed using Sparky. The signals in

the $^1$H-$^{15}$N HSQC spectra were assigned based on the database information (BMRB: 27628) as well as the above NMR data.

**Docking simulation**. To make the binding model of KPMF-8 or resveratrol with the SIRT1 NTD visually clear, a docking simulation analysis was performed by using the AutoDock Vina program[30]. The SIRT1 NTD segment in the SIRT1 X-ray crystallography structure (PDB: 4ZZH) was used as a receptor in this computational procedure.

**MD simulations**. MD simulations were performed starting from the models of complexes SIRT1 NTD–KPMF-8 and SIRT1 NTD–resveratrol generated by docking simulations. The initial structure of each model was prepared as follows. C-terminus was blocked by an $N$-methyl group and histidine residue of SIRT1 NTD (H191) was protonated on the N$^{\varepsilon2}$ atom. The model was then immersed in a cubic box of water, ensuring a minimum distance of 10 Å between any box face and any protein atom. Potassium ions were added to neutralize the system. Amber ff14SB force field parameters[35] were used for energy minimization of the protein molecules, GAFF2[36] was used for the ligand, and the TIP3P model[37] was used for water. After energy minimization and equilibration, 1-μs production MD runs were performed three times with different initial velocities. During the simulation, the temperature was kept at 300 K using the velocity-rescaling method[38], and the pressure was kept at $1.0 \times 10^5$ Pa using the Berendsen weak coupling method[39]. Bond lengths involving hydrogen atoms were constrained using LINCS algorithm[40,41] to allow the use of a large time step (2 fs). Electrostatic interactions were calculated using the particle mesh Ewald method[42,43]. All MD simulations were performed with Gromacs 2019[44], with coordinates recorded every 10 ps. Cluster analysis was performed as follows[45]. RMSD between protein Cα atoms and ligand non-hydrogen atoms was used as the measure of the distance between two structures. Firstly, a structure was taken from the trajectories and was used as a tentative reference structure. Secondly, structures within the cut-off of 2.0 Å were superposed on the reference structure. Thirdly, the average structure was calculated from the superposed structures. The second and the third steps were repeated using the average structure of the previous cycle as the reference structure. When the RMSD between the average structure of the present cycle and that of the previous cycle became less than 0.01 Å, the structures within the cut-off were assigned to a new cluster and were removed from the trajectories. The first snapshot structure of the first run was used as the tentative reference structure at the beginning and these procedures were repeated until the trajectories became empty.

**FRET assay**. SIRT1 (183–510+GS+641–665) was inserted between two fluorescence proteins, Clover and mRuby2. Clover was conjoined to the N-terminus of SIRT1 as a donor, and mRuby2 was conjoined to the C-terminus of SIRT1 as an acceptor. Then this recombination cDNA was infused to vector pBAD. Supplementary Fig. 5 describes the donor and acceptor used when performing the FRET assay in this study. The verified recombinant plasmid was used to transform $E.\ coli$ strain BL21(DE3) (Novagen). The target recombinant protein was overexpressed and purified with the same procedure use for SIRT1 (183–510+GS+641–665) expression and purification. The purity of all protein was confirmed by SDS-PAGE. Its concentration was determined by its absorbance at 280 nm with a Nanodrop ND-1000 (Thermo Fisher Scientific), using its molar extinction coefficient, and calculated using the XtalPred server.

The donor's excitation wavelength (505 nm) was used as the excitation wave, and fluorescent intensities at 515 nm and 600 nm were recorded using a fluorescence spectrophotometer. The FRET ratio (Int (mRuby2)/Int (Clover)) was calculated from the intensity of the acceptor at acceptor emission and the donor at donor emission.

**HDAC activity assay**. An HDAC Fluorometric Cellular Activity Assay BML-AK503 (Enzo Life Sciences) was used to determine HDAC activity in MCF-7 cells according to the manufacturer's instructions. MCF-7 cells were seeded in 100 μl MEM media per well at $2 \times 10^4$ cells/well and grown for 12 h at 37 °C. Subsequently, the media were replaced with 50 μl/well of media containing 0.2 mM FdL substrate with or without an activator for 2 h at 37 °C. To terminate the deacetylation process and develop the fluorescence signal, 50 μl/well of the developer solution and 2 μM Trichostatin A (TSA), dissolved in a lysis buffer, were added to stop the deacetylation process. Plates were incubated for an additional 30 min at 37 °C, and fluorescence (Ex. 360 nm, Em. 460 nm) was measured using a fluorescence plate reader SpectraMax i3. A "time zero" experiment was also performed according to the manufacturer's instruction. Unless otherwise indicated, all measurements were means of four replicates. KPMF-8 and resveratrol were dissolved at 10 mM in 50% ethanol and then diluted to 20 μM in media on the day of the assay.

**Knockdown of SIRT1**. SIRT1 knockdown was achieved using siRNA mediated RNA interference. To ensure maximal knockdown, the negative control siRNA (NC) and three separate siRNAs for SIRT1 protein were tested (siRNA1: 5′-UC AGGUAGUUCCUCGAUGUdtdt-3′, siRNA2: 5′-GUAGGCGGCUUGAUGGUA Adtdt-3′, siRNA3: 5′-ACCGCUUGCUAUCAUGAAAdtdt-3′) (Nippon Gene, Tokyo, Japan). Protocol specifications are as follows: 1.25 μL siRNA was transfected into MCF-7 cells using Lipofectamine RNAiMax transfection reagent based on the manufacturer's instructions (Thermo Fisher Scientific).

**Western blot analysis**. Cellular lysates were prepared using ice-cold lysis buffer (50 mM Tris-HCl, 1 mM EDTA, 1 mM EGTA, 150 mM NaCl, 50 mM NaF and 1% TritonX-100) supplemented with complete protease inhibitor cocktail. Protein measurement was carried out using protein assay BCA kit (Fujifilm Wako Pure Chemical). Equal amounts of proteins (30 μg per lane) were resolved with 12% SDS-PAGE and transferred to PVDF membranes (Bio-Rad, Hercules, CA, USA). The membranes were incubated overnight at 4 °C with primary antibodies (1:1000) against SIRT1 (#2310) and β-actin (#4967) (all from Cell Signaling Technology, Beverly, MA, USA). Afterward, the membranes were incubated with anti-rabbit IgG, HRP-linked secondary antibody (Cell Signaling Technology; 1:10000 dilution). Protein bands were visualized by the enhanced chemiluminescence system according to the manufacturer's instructions (Cell Signaling Technology) using a CCD camera.

**Statistics and reproducibility**. The SIRT1 activity assay and FRET assay were performed using biological triplicates or tetraplicates, where the statistical significance was analyzed by one-way ANOVA followed by a Tukey HSD (honestly significant difference) post-hoc test using R program (Version 3.6.0)[46] $P$ values ≤ 0.05 were considered to be statistically significant. The R codes used were provided in Supplementary Data 1.

**Reporting summary**. Further information on research design is available in the Nature Research Reporting Summary linked to this article.

## Data availability

All data relevant to this study are supplied in the manuscript and supplementary files or are available from the corresponding author upon reasonable request. The source data underlying the graphs in figures are provided in Supplementary Data 2. The atomic coordinates data are provided in Supplementary Data 3. The unprocessed gel blot images with size markers are provided in Supplementary Fig. 6.

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

## Acknowledgements

We thank Dr. Saori Kosono for generously allowing us to use the fluorescence plate reader. This work was supported in part by MEXT KAKENHI Grant Numbers JP18H02151 (Grant-in-Aid for Scientific Research (B) to K.N. and M.S.), JP19H03045 (Grant-in-Aid for Scientific Research (B) to M.S. and K.N.), and JP19H05771 (Grant-in-Aid for Scientific Research on Innovative Areas IBmS to M.S. and K.N.) from the Ministry of Education, Culture, Sport, Science and Technology (MEXT) of Japan, and by Platform Project for Supporting Drug Discovery and Life Science Research (Basis for Supporting Innovative Drug Discovery and Life Science Research (BINDS)) from AMED to T.T. and K.S. under Grant Number JP20am0101107 (support number 2610).

## Author contributions

M.Z., Y.L., N.S., J.Y., T.I., J.T. and K.N. conceived the idea for the project. T.I. and J.T. prepared synthetic KPMF-8. M.Z. conducted the biochemical experiments, analyzed the results and prepared the manuscript. T.T. and K.S. performed MD simulations and analyzed the data. P.L., H.F., M.S., K.I., Y.K., Y.L., K.O., M.S., T.A., S.T., F.H., and K.N. assisted in the experiments. M.Z., P.L., T.T., M.S., K.S., F.H., S.T., N.S., J.Y., T.I., J.T. and K.N. discussed in the preparation of the manuscript.

## Competing interests

The authors declare no competing interests.
