## [Peer Review File · Communications Biology]

Reviewers' comments:

Reviewer #1 (Remarks to the Author):

Dear authors,

Thanks for your detail investigation and gradually pursuing of the mechanism of new STAC which you announced, you tried to answer to some unclear question about the mechanism of KPMF-8 on sirt1 with evaluating of some technique such as ITC, FRET, NMR and...all the methods are agreeable, But with regard to my experience and interest in the STAC biology study, I suggest to best introduce your results and strength speaking with mechanism base, performing molecular dynamic method is left in this manuscript. with MD you can get DGbinding, radius gyration, RMSD and RMSF which can better justify the experimental results.

overall, this manuscript quality is good, results is moderate, but if MD analysis is performed, it will be excellent, the writing is good, discussion is excellent with comparable study,

taken together, the publish of this study in communication biology is agreeable, but if the authors do MD analysis, it will be excellent.

Reviewer #2 (Remarks to the Author):

I thoroughly enjoyed reading this manuscript. It provides novel data, original approaches, and convincing evidence that will help lay to rest a major debate in biology. As such it will be highly regarded and well cited. But first, the cell culture work needs to be bolstered among other improvements.

SIRT1 is an NAD⁺-dependent deacetylase that has been shown in mouse studies, and now in numerous human clinical trials, to protect against age-related diseases, including type 2 diabetes, kidney damage, fatty liver and inflammation. The SIRT1 activator field was almost derailed by an acrimonious debate over the initial use of fluorophore-tagged substrates. At the time, STACs seemed to increase SIRT1 activity only toward fluorophore-tagged substrates but not toward nontagged peptides. As such, there was considerable debate whether the STACs acted directly on SIRT1.

A few years ago, SIRT1 was found to be activated by polymethoxyflavonoids from the Thai black ginger *Kaempferia parviflora*, the most potent of which was quercetin 3,5,7,3',4'-pentamethyl ether (KPMF-8). The apparent 50-fold increase in potency over resveratrol, the previous record holder, is a stunning result that is confirmed and extended in this study. This study has allayed my concerns about this result, and substantially extends our knowledge about this and other STACs. No doubt other labs will begin testing this molecule in mouse and possibly human studies after this study has shown its mechanism of action and shown it to be a STAC that works through direct binding to SIRT1.

Through a rigorous and logical series of biochemical experiments with un-tagged peptides, the authors show that resveratrol and KPMF-8 both bind to the SIRT1 N-terminus (with KPMF-8 binding more tightly than resveratrol), and both do so more tightly in the presence of Ac-p53 peptide. This suggests that SIRT1 binding to a substrate causes a conformational change that exposes the allosteric binding site to its activator. The proposed activation mechanism - the binding of the STAC to SIRT1 assists substrate binding by lowering the K_D value of SIRT1 [E] and Ac-p53 peptide [S] fits with the data. These results, combined with previous data for the activator SRT1460, helps explain why the degree of activation is dependent on the amino-acid sequence of the substrate that Hubbard et al. reported in Science.

To date, an understanding of the binding of STACs to SIRT1 has been facilitated by solving crystal structures but these have obvious drawbacks including potential artifacts of crystallization. The FRET SIRT1 N- and C-termini experiment is original and works brilliantly, potentially opening up a way to measure activation in real-time (see suggestion). The authors are also to be commended for succeeding in generating NMR structures of resveratrol and KMPF-8 bound to the N-terminus in a solution state. This has been one of the holy grails of the field and it not an easy experiment because resveratrol binds weakly. Suggestions to improve the manuscript are below, with emphasis on the cell culture work. The use of statistics is appropriate.

Main points:

1. The manuscript says "The crystal structures of STAC1-SIRT1 and resveratrol-SIRT1-Ac-p53-AMC complexes (PDB: 4ZZH and 5BTR, respectively) provide unambiguous visual proof of the direct activation of SIRT1 by small molecules" but it is important to talk about the work of Hubbard et al. Science (2013) which showed activation can be blocked by a single amino acid change, and Dai et al. Nat. Communications (2015) which has over a dozen crystal structures with a synthetic STAC (and the mutation could be suppressed by another mutation that restored the salt-bridge with the N terminus). Mentioning the mutation will be important for the next experiment suggested in #2.
2. The authors should test if KMPF-8 is blocked in vitro and in vivo by a SIRT1-E230K mutation or equivalent mutation that is known to prevent activation (see Dai 2015, Hubbard et al., 2013). This is the best (and currently the only) way to test if direct activation of SIRT1 by KMPF-8 is occurring in vivo. This is crucial because E230 did not show an obvious chemical shift perturbation in the NMR data, conflicting with the data in Dai and Hubbard.
3. KMPF-8 stimulated SIRT1 deacetylase activity at about fifty times lower concentrations than resveratrol. The differences in activation are highly surprising. Could this be due to oxidation of resveratrol, its conversion from the trans to the cis form, or insolubility? And even though these "results are consistent with" Nakata et al., it would be good if the results were directly compared to those of Hubbard et al. (2013). Are these values in line with that paper?
4. Could the FRET construct be used in cells to detect activation in real-time? The E230K mutation could validate the readout by serving as the perfect negative control. This would be a major breakthrough for the field if it worked.
5. The cell culture experiments are the weakness in the study. The HDAC Fluorometric Cellular Activity Assay from Enzo does not distinguish between SIRT1 and other sirtuins, for example. Because the assay uses crude extracts, it is fraught with potential errors. This is why the mutation experiments in #2 and the suggestion in #4 are important. At a minimum, the experiments need negative controls, e.g. SIRT1 knockouts using CRISPR/Cas9, an E230K mutant or N-terminal deletion, and/or an sh-RNA knockdown of SIRT1. It would also be useful to measure an outcome of SIRT1 activation in the cells, e.g. p53 acetylation, mitochondrial activity, and/or ATP levels.

Minor points:

6. "SIRT1 has been shown to play a critical function in the prevention of many age-related diseases, such as type 2 diabetes, neurodegeneration, cancer, and Alzheimer's disease." Please make sure this is factual. Is it human or is it mouse data the authors are referring to?
7. It would be helpful to point out how the NMR data fits or conflicts with current data (e.g. Dai et al.

and Steegborn et al data).

8. "[STACs] were reported to have limited activity and specificity." Was the specificity limited? They were only SIRT1 activators. Also, resveratrol had 13X activation, which by any measure is a big increase. In the results or discussion, it would be important to state why resveratrol is less active in this assay.

9. It would be helpful to list other known activators on line 55-56. How many are known? How many are known to be direct? Also it would be good to say why the controversy abated here rather than this uninformative sentence: "This challenge spanned several years and led to the discovery of other SIRT1 activators."

10. I would rephrase this sentence: "It would be quite inspiring if this new natural SIRT1 activator were to hold a promising pharmaceutical profile." "Inspiring" is perhaps not the right word here.

11. The introduction ends with " Compared with SIRT1 activation in a non-cellular system, KPMF-8 shows lower but still significant intracellular deacetylase activation in MCF-7 cells." This is expected for small molecules, and I don't think it needs to be pointed out.

12. "To measure the SIRT1 conformational change, we infused Clover" Do you mean integrated or tagged?

13. The authors could discuss their findings in light of xenohormesis - the idea that plant molecules trigger mammalian defenses. Either KPMF-8 looks like an endogenous STAC or we have evolved to sense adversity in the plant world.

Reviewer #1

Dear reviewer:

Thank you for your valuable comments and constructive suggestions. We have revised the manuscript according to your suggestions. The revised portions of the text in the new version of the manuscript are marked in red.

1. Dear authors,

Thanks for your detail investigation and gradually pursuing of the mechanism of new STAC which you announced, you tried to answer to some unclear question about the mechanism of KPMF-8 on sirt1 with evaluating of some technique such as ITC, FRET, NMR and...all the methods are agreeable, But with regard to my experience and interest in the STAC biology study, I suggest to best introduce your results and strength speaking with mechanism base, performing molecular dynamic method is left in this manuscript. with MD you can get DGbinding, radius gyration, RMSD and RMSF which can better justify the experimental results. overall, this manuscript quality is good, results is moderate, but if MD analysis is performed, it will be excellent, the writing is good, discussion is excellent with comparable study, taken together, the publish of this study in communication biology is agreeable, but if the authors do MD analysis, it will be excellent.

[Reply]

We have performed MD analysis, obtained RMSD values, and calculated the contact probabilities of each amino acid to justify the data obtained using NMR and simulated binding model results. DG binding and radius of gyration were not reliably calculated, probably due to weak binding. We have added this information in the revised manuscript (page 11 line 249 – page 12 line 272; page 22 line 505 – page 23 line 521; page 29 line 662 - page 30 line 687; page 33 lines 753 – 757; Fig. 7; Supplementary Fig. 4).

Reviewer #2

Dear reviewer:

Thank you for your valuable comments and constructive suggestions. We have revised the manuscript according to your suggestions. The revised portions of the text in the new version of the manuscript are marked in red.

Main points:

1. The manuscript says "The crystal structures of STAC1–SIRT1 and resveratrol–SIRT1–Ac-p53-AMC complexes (PDB: 4ZZH and 5BTR, respectively) provide unambiguous visual proof of the direct activation of SIRT1 by small molecules" but it is important to talk about the work of Hubbard et al. *Science* (2013) which showed activation can be blocked by a single amino acid change, and Dai et al. *Nat. Communications* (2015) which has over a dozen crystal structures with a synthetic STAC (and the mutation could be suppressed by another mutation that restored the salt-bridge with the N terminus). Mentioning the mutation will be important for the next experiment suggested in #2.

[Reply]

The work of Hubbard *et al.* (2013), Ref. 18, is remarkable and noteworthy. We have included the above reference both in our original and revised manuscripts (Ref. 18) and mentioned the importance of residue E230 in the Introduction section (page 4 lines 78 – 81).

2. The authors should test if KPMF-8 is blocked *in vitro* and *in vivo* by a SIRT1-E230K mutation or equivalent mutation that is known to prevent activation (see Dai 2015, Hubbard et al., 2013). This is the best (and currently the only) way to test if direct activation of SIRT1 by KPMF-8 is occurring *in vivo*. This is crucial because E230 did not show an obvious chemical shift perturbation in the NMR data, conflicting with the data in Dai and Hubbard.

[Reply]

E230 showed a moderate chemical shift perturbation in the SIRT1 NTD–KPMF-8 NMR data but did not display a significant chemical shift perturbation in the SIRT1 NTD–resveratrol NMR data.

To examine the importance of the NTD and E230 for KPMF-8/resveratrol-induced activation *in vitro*, we have additionally measured deacetylation activity of the SIRT1 NTD-deletion construct (243-510) and SIRT1 E230A mutant in the absence or presence of KPMF-8/resveratrol using the FdL SIRT1 fluorimetric drug discovery

assay kit (Enzo Life Sciences, Inc.). In the absence of the activator, the SIRT1 NTD-deletion construct and SIRT1 E230A mutant had comparable activities (76% and 95%, respectively) with full-length SIRT1 included in the kit (Supplementary Fig. 1B). Then, we compared the deacetylation activities in the absence or presence of KPMF-8/resveratrol. The deacetylase activity of SIRT1 NTD-deletion construct was completely insensitive to KPMF-8 or resveratrol (Supplementary Fig. 1C) and that the deacetylase activity of SIRT1 E230A mutant dramatically attenuated sensitivity to KPMF-8 or resveratrol (Supplementary Fig. 1D) compared with full-length SIRT1. Thus, SIRT1 activation effect of KPMF-8 depends on the NTD of the protein and primarily on the residue E230, as is the case with other STACs including resveratrol (Hubbard *et al.*, 2013, Dai *et al.*, 2015) (Refs. 18, 14) (page 6 lines 121 – 132).

Moreover, we also used HPLC-mass spectrometry analysis to indicate direct activation of SIRT1 by KPMF-8 and resveratrol *in vitro*. This assay is based on the O-Acetyl ADP ribose mass spectrometry assay (OAcADPR assay) shown in Hubbard *et al.* (2013) (Ref. 18), with minor modifications. Deacetylation of Ac-p53 peptide catalyzed by SIRT1 was directly activated by KPMF-8 or resveratrol (Supplementary Fig. 2) (page 6 line 133 – page 7 line 143).

In cell SIRT1 activation by KPMF-8 has been examined in a SIRT1 knockdown experiment, which is described in detail below in our reply to your Comment 5.

3. KPMF-8 stimulated SIRT1 deacetylase activity at about fifty times lower concentrations than resveratrol. The differences in activation are highly surprising. Could this be due to oxidation of resveratrol, its conversion from the trans to the cis form, or insolubility? And even though these "results are consistent with" Nakata *et al.*, it would be good if the results were directly compared to those of Hubbard *et al.* (2013). Are these values in line with that paper?

[Reply]

The differences in activation between KPMF-8 and resveratrol are indeed highly surprising. The degree of activation by resveratrol in our work (around 20-fold at 100 μ M) was higher than the value obtained during the screening for mutations affecting SIRT1 activation assay (around 3-fold at 100 μ M), performed by Hubbard *et al.* (2013) (Ref. 18). The difference in activation values may be attributed to the different assay methods used. We have used FdL SIRT1 fluorimetric drug discovery assay (Enzo), whereas Hubbard *et al.* (2013) (Ref. 18) used BIOMOL assay, which was a smarter and more scientifically rigorous assay because all their reactions were normalized to control reactions in the absence of β -NAD. However, our high

values for activation by resveratrol were reproducible and in line with the results of Nakata *et al.* (2014) and Kaeberlein *et al.* (2005) (around 20-fold at 100 μ M) (Refs. 19, 9).

Resveratrol used by us is sold as a reagent (Fujifilm Wako Pure Chemical, Osaka, Japan) and was properly stored prior to the assay. Thus, we do not believe that the differences in activation were due to the oxidation of resveratrol, its conversion from *trans* to *cis* form, or its insolubility.

4. Could the FRET construct be used in cells to detect activation in real-time? The E230K mutation could validate the readout by serving as the perfect negative control. This would be a major breakthrough for the field if it worked.

[Reply]

It would be indeed remarkable if the FRET construct could be used to detect the activation in real time inside cells. However, we believe that it would be difficult to conduct *in vivo* FRET experiments since we do not have the equipment to detect *in vivo* FRET. Also, since the main objective of our current work was to elucidate whether KPMF-8 activates SIRT1 activity directly, we will attempt to detect *in vivo* FRET in our future studies.

5. The cell culture experiments are the weakness in the study. The HDAC Fluorometric Cellular Activity Assay from Enzo does not distinguish between SIRT1 and other sirtuins, for example. Because the assay uses crude extracts, it is fraught with potential errors. This is why the mutation experiments in #2 and the suggestion in #4 are important. At a minimum, the experiments need negative controls, e.g. SIRT1 knockouts using CRISPR/Cas9, an E230K mutant or N-terminal deletion, and/or an sh-RNA knockdown of SIRT1. It would also be useful to measure an outcome of SIRT1 activation in the cells, e.g. p53 acetylation, mitochondrial activity, and/or ATP levels.

[Reply]

To distinguish SIRT1 from other sirtuins and deacetylases, we performed a SIRT1 knockdown experiment and analyzed the deacetylase activity of SIRT1 knockdown MCF-7 cells after the addition of KPMF-8 or resveratrol into the medium. Deacetylase activation by KPMF-8 and resveratrol occurred in normal MCF-7 cells (Fig. 9A, as shown in the original manuscript) but not in SIRT1 knockdown MCF-7 cells (Fig. 9C, D), and these activators did not affect the expression levels of SIRT1 (Fig. 9B), supporting the hypothesis that SIRT1 is the target deacetylase molecule

that is activated by KPMF-8 and resveratrol. We have added related statements explaining this in the revised manuscript, which are marked in red (page 14 lines 311 – 323; page 24 line 553 – page 25 574; page 34 lines 764 – 771).

Minor points:

6. "SIRT1 has been shown to play a critical function in the prevention of many age-related diseases, such as type 2 diabetes, neurodegeneration, cancer, and Alzheimer's disease." Please make sure this is factual. Is it human or is it mouse data the authors are referring to?

[Reply]

With regards to previous research conducted on SIRT1 for the prevention of many age-related diseases, we have only referred to mouse data. We have made this distinction clear in our revised manuscript (page 3 lines 38 – 40).

7. It would be helpful to point out how the NMR data fits or conflicts with current data (e.g. Dai et al. and Steegborn et al data).

[Reply]

We have added discussion on our NMR data, comparing it with the data reported by Dai *et al.* and Steegborn *et al.* (page 15 line 356 – page 16 line 363).

8. "[STACs] were reported to have limited activity and specificity." Was the specificity limited? They were only SIRT1 activators. Also, resveratrol had 13X activation, which by any measure is a big increase. In the results or discussion, it would be important to state why resveratrol is less active in this assay.

[Reply]

Here, we want to point out that some STACs, such as resveratrol, increase the lifespan of yeasts (Howitz *et al.* 2003. *Nature*) (Ref. 6), as well as prolong the lifespan of metazoans (Wood *et al.* 2004. *Nature*) (Ref. 7) and mice (Baur *et al.* 2006. *Nature*) (Ref. 8) by simultaneously acting on several target proteins and not just by modulating SIRT1 activity. Moreover, studies suggesting the effect of resveratrol and similar polyphenols on insulin sensitivity and stress resistance in animal models are limited, probably due to their poor systemic bioavailability and rapid metabolism. SIRT1 catalytic rate stimulation by resveratrol was reported to have a 13.4x increase (Howitz, *et al.* 2003. *Nature*) (Ref. 6). In our research, resveratrol at the

concentration of 100 μ M also showed around 20x activation. The reason why resveratrol was shown to be less active in other publications might be its poor systemic bioavailability and rapid metabolism, as the authors have indicated.

As per the reviewer's suggestion, we have revised this sentence, which was not clear, and marked the revised section in red (page 3 lines 51 – 55).

9. It would be helpful to list other known activators on line 55-56. How many are known? How many are known to be direct? Also it would be good to say why the controversy abated here rather than this uninformative sentence: "This challenge spanned several years and led to the discovery of other SIRT1 activators."

[Reply]

We thank the reviewer for pointing out an inconclusive statement. In the revised manuscript, we listed other known activators and mentioned whether these compounds directly or indirectly activate SIRT1. The revised section is marked in red (page 3 line 56 – page 4 line 65).

10. I would rephrase this sentence: "It would be quite inspiring if this new natural SIRT1 activator were to hold a promising pharmaceutical profile." "Inspiring" is perhaps not the right word here.

[Reply]

Thank you for pointing out this statement. This sentence is revised and marked in red (page 5 lines 92 – 95).

11. The introduction ends with " Compared with SIRT1 activation in a non-cellular system, KPMF-8 shows lower but still significant intracellular deacetylase activation in MCF-7 cells." This is expected for small molecules, and I don't think it needs to be pointed out.

[Reply]

Thank you for your insightful suggestion. We have removed this sentence from the revised manuscript (page 5 line 104).

12. "To measure the SIRT1 conformational change, we infused Clover" Do you mean integrated or tagged?

[Reply]

We tagged Clover and mRuby2 to the SIRT1 N- and C-termini, respectively. This statement has been revised in the manuscript (page 13 line 284).

13. The authors could discuss their findings in light of xenohormesis -the idea that plant molecules trigger mammalian defenses. Either KPMF-8 looks like an endogenous STAC or we have evolved to sense adversity in the plant world.

[Reply]

The use of plant-derived compounds for promoting our health is based on traditional herbal medicine developed in China, India, as well as European countries. Our data indicate the possibility of identifying more useful compounds from the wide variety of plant secondary metabolites, particularly polyphenols, with protective activity against UV damage and oxidation. In this manuscript, we will not include any discussion related to this.

Reviewers' comments:

Reviewer #2 (Remarks to the Author):

The authors have done an excellent job of addressing my concerns and doing additional experiments. The SIRT1 truncation and E230K work is well done and consistent with expectations.

Reviewer #3 (Remarks to the Author):

The research group applied both experimental and simulation methods to prove that KPMF-8 activates SIRT1 more effectively than resveratrol. In the simulation section, first auto-dock was applied to predict the binding structure of SIRT1 NTD with KPMF-8 and resveratrol. After that gromacs all-atom simulations were performed for 1.0 us with 3 repeats using different velocities.

The following information should be implemented in the manuscript:

- 1). How the RMSD was calculated? Is that based on the backbone Ca atoms or based on coordinates of all the atoms? Was RMSD calculated after doing structure alignment? Which is the reference structure in alignment?
- 2). During the clustering, what is the reference structure? Is the clustering based on structure deviation from the reference structure?
- 3). In the RMSD result, the protein structure is stable, but the ligand structure changed significantly especially in the run2 for KPMF-8 and resveratrol which even could reach 20Å. Is that really the RMSD of ligand or the RMSD of the complex of ligand bound with protein? What really happened to the complex with a RMSD of 20 Å and above?

If dissociation happened with a RMSD of 20 Å in simulations, what is the binding affinity between protein and different ligands? The binding interaction energy between protein and different ligands should be calculated and compared to support the observation.

Reviewer #3

Dear reviewer,

Thank you for your valuable comments. The following are point-by-point responses to the comments. We have attached the revised manuscript in which the revised portions of the text are highlighted in yellow.

The research group applied both experimental and simulation methods to prove that KPMF-8 activates SIRT1 more effectively than resveratrol. In the simulation section, first auto-dock was applied to predict the binding structure of SIRT1 NTD with KPMF-8 and resveratrol. After that gromacs all-atom simulations were performed for 1.0 us with 3 repeats using different velocities. The following information should be implemented in the manuscript:

1. How the RMSD was calculated? Is that based on the backbone Ca atoms or based on coordinates of all the atoms? Was RMSD calculated after doing structure alignment? Which is the reference structure in alignment?

[Reply]

RMSD values were calculated for C α atoms of the protein and non-hydrogen atoms of the ligands using the docking models as the reference structures after aligning the C α atoms of the protein of the MD structures to those of the docking models.

We added these descriptions to the legend of Fig. 7 of the revised manuscript.

2. During the clustering, what is the reference structure? Is the clustering based on structure deviation from the reference structure?

[Reply]

Cluster analysis was performed as follows. RMSD between protein C α atoms and ligand non-hydrogen atoms was used as the measure of the distance between two structures. Firstly, a structure was taken from the trajectories and was used as a tentative reference structure. Secondly, structures within the cut-off of 2.0 Å were superposed on the reference structure. Thirdly, the average structure was calculated from the superposed structures. The second and the third steps were repeated using the average structure of the previous cycle as the reference structure. When the RMSD between the average structure of the present cycle and that of the previous cycle became less than 0.01 Å, the structures within the cut-off were assigned to a new cluster and were removed from the trajectories. The first snapshot structure of the first run was used as the tentative reference structure at the beginning and these procedures were repeated until the trajectories became empty.

We added these descriptions to the "MD simulation" section of Method in the revised manuscript.

3. In the RMSD result, the protein structure is stable, but the ligand structure changed significantly

especially in the run2 for KPMF-8 and resveratrol which even could reach 20Å. Is that really the RMSD of ligand or the RMSD of the complex of ligand bound with protein? What really happened to the complex with a RMSD of 20 Å and above? If dissociation happened with a RMSD of 20 Å in simulations, what is the binding affinity between protein and different ligands? The binding interaction energy between protein and different ligands should be calculated and compared to support the observation.

[Reply]

The figure below shows the superposition of the representative structure (cyan) of a cluster with the RMSD values of 21.74 ± 0.57 Å on the SIRT1 NTD–KPMF-8 docking model (brown).

As can be seen here, when the ligand binds to the back side of the protein, the RMSD value becomes more than 20 Å. Although this cluster is the largest among the cluster with the average RMSD being more than 20 Å, its fraction of the population is only 1.99 %. To illustrate the fact that large-RMSD binding modes rarely occur, we calculated probability distributions as a function of the RMSD. As shown in the figure below, the probabilities of around 20 Å are very small in both the plots for SIRT1 NTD–KPMF-8 and SIRT1 NTD–resveratrol. The sums of the probabilities for $\text{RMSD} > 12$ Å were 15.7% and 9.74% for SIRT1 NTD–KPMF-8 and SIRT1 NTD–resveratrol, respectively. Therefore, these modes little affected the NMR chemical shifts and the binding modes with large RMSD values are ignorable.

On the other hand, the large peaks in the region of $\text{RMSD} < 10$ Å come from the large clusters

shown in Figure S4. For SIRT1 NTD–KPMF-8, the largest four clusters with the fractions of the population of 16.3%, 10.1%, 10.0%, and 9.9% had RMSD values of 7.93 ± 0.88 Å, 4.29 ± 1.37 Å, 7.70 ± 0.42 Å, and 5.09 ± 0.61 Å (average \pm standard deviation). For SIRT1 NTD–resveratrol, the RMSD values of the largest three clusters (28.4%, 22.9%, and 10.1%) were 5.08 ± 1.64 Å, 8.06 ± 0.64 Å, and 8.91 ± 0.93 Å. These results indicate that frequently occurring binding modes have relatively small RMSD values.

The free-energy difference between different binding states is related with the probabilities of existence of these states. Let p_i be the probability of existence of state i . The free-energy difference between binding states 1 and 2 is given by,

$$\Delta G = G_i - G_j = -RT \ln \frac{p_i}{p_j}$$

Therefore, the binding modes shown in Figure S4 are more stable than those with large RMSD values.

To show that the large clusters shown in Figure S4 have relatively small RMSD values, we added RMSD values (average \pm standard deviation) to Figure S4. In the previous manuscript, we erroneously showed the results of the cluster analysis performed only for the trajectory of run 1. We therefore replaced the structural images with those obtained from the cluster analysis performed for the trajectories of all the runs.

REVIEWERS' COMMENTS:

Reviewer #3 (Remarks to the Author):

The revised work is acceptable.